# PATCH-LEVEL KERNEL ALIGNMENT FOR DENSE SELF-SUPERVISED LEARNING

## ABSTRACT

Dense self-supervised learning (SSL) methods showed its effectiveness in enhancing the fine-grained semantic understandings of vision models. However, existing approaches often rely on parametric assumptions or complex post-processing (e.g., clustering, sorting), limiting their flexibility and stability. To overcome these limitations, we introduce Patch-level Kernel Alignment (PaKA), a non-parametric, kernel-based approach that improves the dense representations of pretrained vision encoders with a post-(pre)training. Our method propose a robust and effective alignment objective that captures statistical dependencies which matches the intrinsic structure of high-dimensional dense feature distributions. In addition, we revisit the augmentation strategies inherited from image-level SSL and propose a refined augmentation strategy for dense SSL. Our framework improves dense representations by conducting a lightweight post-training stage on top of a pretrained model. With only 14 hours of additional training on a single GPU, our method achieves state-of-the-art performance across a range of dense vision benchmarks, demonstrating both efficiency and effectiveness.

## 1 INTRODUCTION

Self-supervised learning (SSL) has rapidly advanced the capabilities of vision foundation models, enabling them to learn generalizable visual concepts without human-annotated labels. While earlier work (Chen et al., 2020; He et al., 2020; Grill et al., 2020; Caron et al., 2020) primarily focused on image-level tasks such as classification, recent studies have shifted toward fine-grained, structured tasks, such as semantic segmentation (Strudel et al., 2021; Xie et al., 2021; Zhang et al., 2022) and object detection (Carion et al., 2020; Li et al., 2022a) that require detailed, spatially-aware understanding. Achieving strong performance in such tasks hinges on the quality of dense representations, where each spatial location in the image is embedded with semantically meaningful information.

To enable such dense visual understanding, recent research has shifted towards dense self-supervised learning, aiming to equip models with fine-grained spatial awareness. Some approaches (Oquab et al., 2023; Zhou et al., 2022) have advanced dense representation learning by reconstructing masked image patches, which result in high-quality, fine-grained features by leveraging the input itself as the target. Meanwhile, another line of research (Lebailly et al., 2024; Pariza et al., 2025; Stegmüller et al., 2023; Ziegler & Asano, 2022) has focused on explicitly aligning teacher and student patch-level features using carefully designed objectives without masking. Specifically, methods like Leopart (Ziegler & Asano, 2022), Croc (Stegmüller et al., 2023), and CrIBO (Lebailly et al., 2024) apply clustering algorithms to group features and train models using clustering-based pseudo-labels. On the other hand, NeCo (Pariza et al., 2025) adopts a sorting-based objective to enforce patch-level relational consistency and builds on pretrained encoders. This approach exemplifies what we refer to as post-(pre)training, where a pretrained model is taken and further refined for enhanced dense representations.

At their core, these methods can be seen as performing distribution alignment, where alignment occurs at the level of dense feature representations. However, they often rely on parametric assumptions to manage the complexity of patch-level feature distributions in high-dimensional space. For instance, cluster-based approaches (Lebailly et al., 2024; Stegmüller et al., 2023) model the probability distribution of a patch by mapping it to $K$ predefined prototypes. Such parametric assumptions can make training sensitive to hyperparameter choices, while also reducing the flexibility to capture the complex distributions of patch-level features.

In this work, we propose *a non-parametric, kernel-based learning approach that moves beyond the limitations of parametric distribution modeling, formulated as a post-(pre)training refinement stage applied to enhance the dense representations* of a pretrained model. Our method evaluates the holistic similarity structure of patch-level features through a kernel metric. While a simple Gram matrix alignment can be used to compare teacher and student patch features, we find this approach unstable due to structural misalignment between teacher and student feature spaces in dense post-(pre)training. To address this, we introduce Patch-level Kernel Alignment (PaKA), which applies Centered Kernel Alignment (CKA) at the patch level to capture intrinsic similarity structures, enabling reliable comparison under distributional discrepancies. Consequently, PaKA enables stable and efficient dense post-(pre)training without parametric constraints and eliminates the need for complex algorithms or memory banks.

Furthermore, we revisit augmentation strategies originally designed for image-level SSL, which have been widely adopted in dense SSL (Pariza et al., 2025; Ziegler & Asano, 2022) without thorough re-evaluation. Our analysis shows that certain augmentations, especially those inherited from image-level SSL, can hinder the learning of spatially detailed dense features. Motivated by these observations, we propose a cropping strategy specifically designed to preserve spatial information and improve dense feature alignment.

Our framework, combining CKA-based alignment and refined augmentation, achieves state-of-the-art performance on multiple dense benchmarks while reducing computation by 37% and memory usage by 24% compared with the prior post-training method NeCo (Pariza et al., 2025). The main contributions of our work are summarized as follows:

- We propose Patch-level Kernel Alignment (PaKA), a simple yet effective dense post-(pre)training method that aligns dense features between teacher and student models using Centered Kernel Alignment (CKA), without relying on clustering, memory banks, or explicit distribution modeling.

- We analyze the impact of standard augmentation strategies inherited from image-level SSL and identify their limitations for dense representation learning. Based on this analysis, we introduce a new augmentation strategy specifically tailored for dense representation learning.

- We demonstrate that our full framework, which combines CKA-based alignment and refined augmentations, achieves state-of-the-art performance across diverse dense vision benchmarks while reducing computational and memory costs.

## 2 RETHINKING DISTRIBUTION ALIGNMENT FOR DENSE SSL

### 2.1 DISTRIBUTION ALIGNMENT IN DENSE SELF-SUPERVISED LEARNING

The process of dense self-supervised learning (SSL) can be interpreted as a problem of distribution alignment. At its core, dense SSL leverages a self-distillation mechanism to learn fine-grained representations by enforcing multi-view consistency at the patch level. This is commonly implemented through a student-teacher architecture (Caron et al., 2021), where the teacher network generates a target distribution of dense representations from a holistic, global view of an image. This teacher distribution acts as a stable, information-rich target, encapsulating the underlying semantic structure. The student network, conversely, is tasked with modeling this target distribution while only observing partial, low-resolution local views. By aligning the student's output distribution with the teacher's richer target distribution, the dense SSL objective guides the student to produce robust and spatially-aware representations, even from partial views.

### 2.2 FROM PARAMETRIC CONSTRAINTS TO NON-PARAMETRIC RELATIONAL LEARNING

Despite their diverse formulations, contemporary dense self-supervised learning approaches (Wang et al., 2021; Stegmüller et al., 2023; Lebailly et al., 2024; Ziegler & Asano, 2022; Pariza et al., 2025) can be seen to effectively perform distribution alignment in feature space. However, these approaches are often built upon parametric assumptions and architectural priors, which limit their flexibility in learning the complex and high-dimensional distribution of patch-level features.

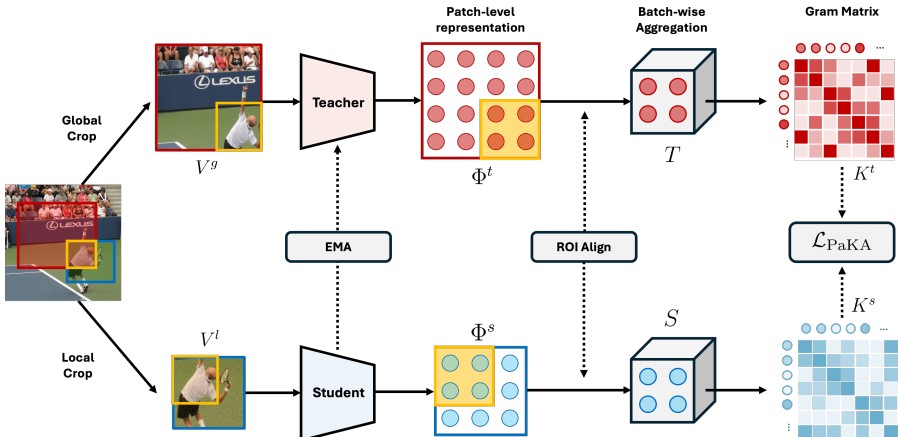

Figure 1: **Overview of Patch-level Kernel Alignment (PaKA) for Dense Post-(pre)training.** PaKA is a student-teacher framework that aligns dense patch representations by comparing their relational structures, enabling the student to capture the teacher's fine-grained feature relationships without requiring complex algorithms or memory banks. Our post-training backbone is DINOv2R (ViT-S/14).

For instance, DenseCL (Wang et al., 2021) learns patch-level representations using contrastive learning. This method defines a probability distribution for patches under a binary constraint - whether a patch is similar (positive) or dissimilar (negative) to another - and it only leverages these binary relationships during training. Such a limited definition can restrict the diversity of patch relationships captured in the learned distribution.

Cluster-based approaches (Lebailly et al., 2024; Stegmüller et al., 2023; Ziegler & Asano, 2022) attempt to partially overcome such limitations by defining the probability distribution of a patch through comparison against $K$ prototypes, often generated across multiple images. Nevertheless, this approach remains parametric, as it relies on pre-defined parameters such as the number of clusters $K$, which constrains the flexibility of the learned distribution.

Similarly, NeCo (Pariza et al., 2025), whose approach of matching relationships among patches within a batch inherently avoids the need for explicit prototype construction, nonetheless introduces its own form of restriction on the distribution. It establishes a soft order of similarity among patches via a sorting mechanism with a steepness parameter $s$. The high steepness decisively sharpens the ranking among patches and drives the output toward a discrete limit. While this strategy circumvents the need for explicit clustering, the sorting process implicitly induces a structure that resembles cluster-based grouping, where top-ranked patches act as pseudo-prototypes.

To address these limitations, we introduce a non-parametric and relational learning framework without imposing a fixed structure on the feature distribution. This is achieved via a kernel-based mechanism, which enables flexible modeling of complex feature relationships.

## 3 PATCH-LEVEL KERNEL ALIGNMENT

We introduce Patch-level Kernel Alignment (PaKA), a simple and flexible framework for dense SSL, as illustrated in Figure 1. PaKA fine-tunes a pretrained image-level SSL model to learn spatially and semantically rich patch-level representations using a kernel-based alignment loss.

### 3.1 POST-(PRE)TRAINING VISION ENCODERS FOR DENSE REPRESENTATION

Following prior works (Ziegler & Asano, 2022; Pariza et al., 2025), we further fine-tune the image-level SSL models such as DINOv2 (Oquab et al., 2023) to improve patch-level dense representations. Both student and teacher networks are initialized from the same pretrained weights, with the teacher updated via Exponential Moving Average (EMA). We apply a multi-crop augmentation strategy (Caron et al., 2020), generating two global crops $V_g$ for both networks and multiple low resolution local crops $V_l$ exclusively for the student. From the global crop $V_g$, divided into $H \times W$

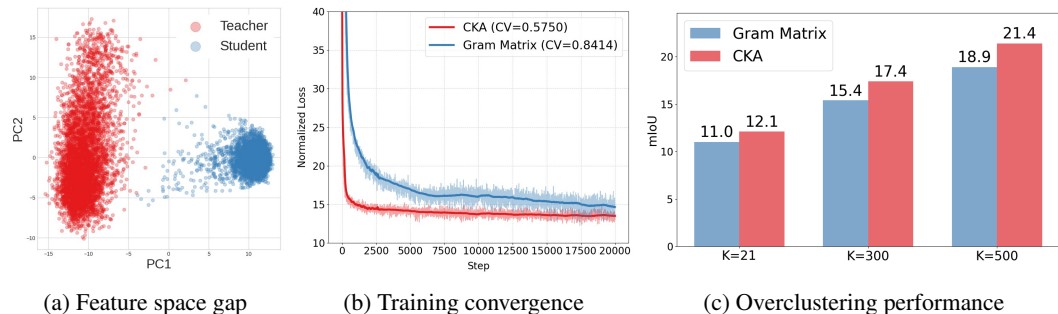

|(a) Feature space gap|(b) Training convergence|(c) Overclustering performance|

Figure 2: **Why CKA leads to better alignment than Gram-matrix in dense post-(pre)training.** (a) 2D PCA projection of 5,000 teacher and student dense representations. (b) Normalized training loss curves for the Gram-matrix and CKA losses. Complete training loss curves can be found in the Appendix. (c) Overclustering performance on ADE20K.

patches, the teacher encoder produces a patch-level dense representation $\Phi^t \in \mathbb{R}^{H \times W \times D}$. Similarly, the student encoder processes a local crop and divides into $h \times w$ patches, to yield its patch-level dense representation $\Phi^s \in \mathbb{R}^{h \times w \times D}$.

To enable a direct patch-wise comparison of $\Phi^t$ and $\Phi^s$, we align representations from the region of intersection between $V_g$ and $V_l$. Specifically, let $b$ be the bounding box defining $V_l$ within $V_g$. We apply ROI Align to the teacher's representation $\Phi^t$ to extract features corresponding to this intersection, resized to a target $h' \times w'$ grid: $\Phi^t_b = \text{ROIAlign}(\Phi^t, b, h', w')$. Similarly, the student's representation $\Phi^s$ is resized to $\Phi^s_b = \text{ROIAlign}(\Phi^s, b, h', w')$. This results in two aligned feature maps, $\Phi^t_b, \Phi^s_b \in \mathbb{R}^{h' \times w' \times D}$, which are subsequently flattened into $N = h' \times w'$ patch embeddings:

$$T = [\mathbf{t}_1, \ldots, \mathbf{t}_N]^\top, \quad S = [\mathbf{s}_1, \ldots, \mathbf{s}_N]^\top \in \mathbb{R}^{N \times D}.$$

The spatially aligned patch embeddings, $T$ and $S$, are dense features that represent the same overlapping region captured from different views. Given the inherently complex and high-dimensional nature of these patch-derived feature sets, $T$ and $S$, it is crucial to adopt an alignment methodology that can effectively capture their intricate relational structures.

### 3.2 KERNEL-BASED RELATIONAL ALIGNMENT

We propose a kernel-based learning objective to align the dense features, $T$ and $S$. This approach is motivated by kernel-based strategies (He & Ozay; Li et al., 2021; Qiu et al., 2024; Zhou et al., 2024; Zong et al.) for knowledge transfer, which facilitate a more holistic comparison by aligning the entire similarity structure of feature distributions.

One intuitive approach to align feature distribution is to directly compare the Gram matrices, $K^t$ and $K^s$, derived from linear kernels on the patch-level features $T$ and $S$.

$$K^t = TT^\top, K^s = SS^\top.$$

These matrices encode the pairwise relationships within each feature set, capturing the fine-grained semantics embedded within the high-dimensional feature space. Indeed, the recent DINOv3 (Siméoni et al., 2025) leverages this concept in its "Gram anchoring" method, which uses a loss equal to $L_{\text{Gram}} = \|K^s - K^t\|_F^2$ to preserve the quality of dense features that would otherwise degrade during long training schedules. While DINOv3 employs this intuitive approach for *preservation*, we first analyze the Gram matrix loss as a baseline to evaluate its effectiveness for representation *improvement*, before introducing our more robust method.

While Gram matrices offer a straightforward means to encode pairwise relationships, directly adopting them in dense post-(pre)training can be suboptimal. Our findings reveal that there is a distributional discrepancy between teacher and student representations. As illustrated in Figure 2a, there is not only a clear spatial gap between the two feature spaces, but also a difference in their geometric structure. This structural disparity is quantitatively evident: the teacher's feature space is highly anisotropic, with a large difference of variances along its first two principal components, in stark contrast to the student's far more compact space. This underlying discrepancy is critical because the Gram matrix,

as a linear kernel, is inherently sensitive to the geometry of its feature space. When the teacher and student spaces are structurally misaligned, their corresponding Gram matrices, $K^t$ and $K^s$, will capture inherently different relational patterns.

To address the challenges due to feature distribution discrepancies, we adopt Centered Kernel Alignment (CKA). It was initially proposed to measure feature similarity across different layers of a neural network, which typically exhibit distinct feature distributions. CKA is theoretically grounded in the Hilbert-Schmidt Independence Criterion (HSIC) (Gretton et al., 2005), allowing it to evaluate the *statistical dependence* between the feature distributions represented by $K^t$ and $K^s$. This enables a comparison of intrinsic similarity structures rather than extrinsic ones, resulting in more robust and efficient training. To quantify this stability, we use the Coefficient of Variation (CV) of each training loss across mini-batches, a statistical measure comparing the relative variability of variables with different scales. Over 20,000 training steps, the CV of the CKA loss is smaller than the CV of Gram matrix loss, indicating that CKA is less sensitive to feature variance and promotes more stable learning. Consequently, as shown in Figure 2b and Figure 2c, this stability facilitates faster convergence, ultimately resulting in higher performance.

### 3.3 FORMULATION OF THE PAKA LOSS

The CKA-based alignment of the global geometry between the patch-level representations $T$ and $S$ is implemented as follows. Starting with the Gram matrices $K^t$ and $K^s$, which are derived using linear kernels as previously defined, we first center them using the centering matrix $H = I - \frac{1}{N}\mathbf{1}\mathbf{1}^\top$:

$$\tilde{K}^s = HK^sH, \quad \tilde{K}^t = HK^tH.$$

The CKA similarity, which quantifies the likeness between the representations $S$ and $T$ using their centered Gram matrices, is then defined as:

$$\text{CKA}(S,T) = \frac{\langle \tilde{K}^s, \tilde{K}^t \rangle_F}{\|\tilde{K}^s\|_F \cdot \|\tilde{K}^t\|_F}.$$

The CKA metric quantifies the dependency between kernels: a high CKA means strong dependency, while a low CKA implies weak dependency. Consequently, for a loss function, we desire a formulation where minimizing the loss corresponds to increasing CKA. As CKA is normalized between 0 and 1, the loss term takes the form of 1 - CKA, effectively converting the maximization of CKA into a minimization problem. The PaKA loss is:

$$\mathcal{L}_{\text{PaKA}} = 1 - \text{CKA}(S,T).$$

Our PaKA loss has a value between 0 and 1 and does not require any hyperparameters. Minimizing this loss encourages the student to preserve the teacher's pairwise patch relationships, offering robust alignment without forcing $\mathbf{s}_i \approx \mathbf{t}_i$ directly.

## 4 TOWARDS EFFECTIVE AUGMENTATION STRATEGIES FOR DENSE SSL

### 4.1 LIMITATIONS OF IMAGE-LEVEL AUGMENTATION FOR DENSE SSL

Data augmentation is pivotal in self-supervised learning (SSL), defining invariances for the model. However, directly applying standard augmentation strategies—originally designed for image-level pre-training—to post-training dense SSL frameworks presents distinct challenges. Image-level SSL methods (Chen et al., 2020; Grill et al., 2020; Caron et al., 2020; 2021) use augmentations (e.g., multi-crop) to create different "views" for instance-level invariance. Dense SSL methods such as NeCo (Pariza et al., 2025) and Leopart (Ziegler & Asano, 2022) retain this paradigm, feeding global crops to both student and teacher, with spatial alignment via ROI Align for local views. This approach has limitations for dense feature alignment:

- *Local crops often have minimal spatial overlap with global crops*, reducing mutual information and alignment utility.

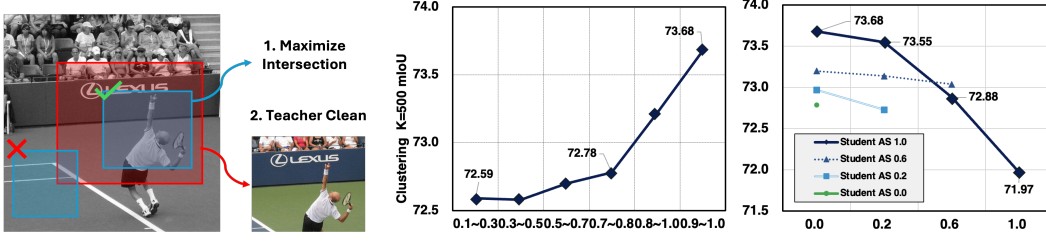

(a) Our augmentation strategies (b) Minimum intersection ratio (c) Teacher augmentation strength

Figure 3: **Our proposed augmentation strategies and their empirical validation.** (a) Conceptual overview of maximizing view intersection and employing a clean teacher. (b) Performance, measured as mIoU in an overclustering task(K=500) on Pascal VOC (Everingham et al.), significantly improves as the minimum intersection ratio between views is increased. (c) Student model performance peaks when teacher augmentation strength is minimized. Detailed experimental results are provided in the Appendix.

- *Strong augmentations on the teacher view may introduce noise*, leading the student to overfit to non-semantic artifacts.

These limitations reduce the mutual information available for patch-level alignment and may introduce noise, limiting the effectiveness of dense SSL.

### 4.2 PROPOSED AUGMENTATION STRATEGIES

Addressing the limitations that arise when standard augmentations are applied to post-training dense SSL, we introduce two key refinements. Our proposed augmentation strategies are depicted schematically in Figure 3a.

**Global–Local Intersection Maximization.** Dense SSL methods (Ziegler & Asano, 2022; Pariza et al., 2025) compute their objective by applying ROI Align exclusively to the overlapping regions from global and local crops of different sizes. A critical issue arises in this process: if the spatial overlap between these global and local crops is minimal, the amount of information incorporated into the loss function diminishes significantly, which can hinder effective learning. To maximize shared information between teacher and student views, we propose Global–Local Intersection Maximization: enforcing a minimum spatial Intersection-over-Union (IoU) between local and global crops. We reject local crops whose IoU with the corresponding global crop is below a threshold ratio $m$. Controlling this minimum overlap $m$ explicitly regulates mutual information. Empirically, increasing $m$ consistently improves clustering performance, as shown in Figure 3b, confirming that denser shared regions lead to more effective alignment. In our method, we set the value of $m$ to ensure an overlap of 90% or greater.

**Reducing Noise with an Augmentation-free Teacher.** Effective dense SSL hinges on capturing fine-grained spatial relationships, not just global semantics. This fundamentally reframes the teacher's role from simply another augmented peer to a stable semantic anchor. We introduce a clean-teacher strategy: the teacher receives an unaugmented or very weakly transformed image, while the student still processes the full augmentation pipeline. Our Clean Teacher Strategy embodies this revised role through an asymmetric configuration: the student learns invariance from strong augmentations while aligning to a teacher that ideally processes an *entirely augmentation-free* input. This allows the teacher to provide more reliable signals reflecting the image's intrinsic structure. Our experiments in Figure 3c support this observation, showing that student representation quality is highest when all augmentations, including color jitter and blur, are removed from the teacher's pipeline.

## 5 EXPERIMENTS

### 5.1 SETUP

**Datasets.** We use the COCO (Lin et al., 2014) dataset for model training, with evaluation conducted across a diverse set of datasets, including Pascal VOC 2012 (Everingham et al.), COCO (Lin et al., 2014), ADE20K (Zhou et al., 2017), and COCO-Stuff 164K (Caesar et al., 2018). These datasets cover various tasks in this study. For visual in-context learning and overclustering, performance is measured using Pascal VOC 2012 and ADE20K. For semantic segmentation, we report linear-probe result on Pascal VOC 2012, COCO-Things, COCO-Stuff 164K, and ADE20K.

**Implementation Details.** Our model is post-trained from the pretrained checkpoint DI-NOv2R (Darcet et al.) with no registers. We conduct the main training on a single H100 GPU with 80GB of memory with a mini-batch size of 55. For a fair comparison, we also post-trained same backbone model by NeCo (Pariza et al., 2025), based on the code from official GitHub repositories. On our hardware environment, NeCo baseline was trained within a maximum batch size of 40. We use AdamW (Loshchilov & Hutter, 2017) optimizer with a learning rate of $5e-6$ for the backbone model, a weight decay of 0.04, and 20 workers. Our models generally underwent 25 epochs of training, while data augmentation experiments were run for 10 epochs. The data augmentation experiments used 4 RTX 3090 GPUs with 24GB of memory in a mini-batch size of 10 per GPU. For crop sizes, we use two $518\times518$ global crops and four $98\times98$ local crops.

### 5.2 COMPARISON WITH PRIOR WORKS

Table 1: **Visual In-Context Learning benchmark.** Dense Nearest Neighbor retrieval performed on Pascal VOC and ADE20k datasets with varying training data proportions. We construct uniformly sampled training subsets, down-sampling the full set by ratios of 1/1, 1/8, 1/64, and 1/128. † reported in this paper are produced using our implementation.

| Method | Pretrained | Pascal VOC | | | | ADE20K | | | |
|---|---|---|---|---|---|---|---|---|---|
| | | 1/128 | 1/64 | 1/8 | 1/1 | 1/128 | 1/64 | 1/8 | 1/1 |
| DINO | ✗ | 26.4 | 30.5 | 41.3 | 48.7 | 9.5 | 11.0 | 15.0 | 17.9 |
| SelfPatch | ✗ | 28.4 | 32.6 | 43.2 | 50.8 | 10.0 | 10.9 | 14.7 | 17.7 |
| CrOC | ✗ | 34.0 | 41.8 | 53.8 | 60.5 | 8.7 | 10.8 | 15.2 | 17.3 |
| Leopart | ✗ | 44.6 | 49.7 | 58.4 | 64.5 | 12.9 | 14.8 | 19.6 | 23.9 |
| CrIBo | ✗ | 53.9 | 59.9 | 66.9 | 72.4 | 14.6 | 17.3 | 22.7 | 26.6 |
| DINOv2R | ✗ | 60.1 | 65.7 | 74.5 | 78.8 | 23.7 | 27.1 | 33.9 | 39.5 |
| NeCo† | DINOv2R | 65.5 | 69.0 | 75.1 | 78.8 | 23.9 | 27.2 | 34.3 | 39.8 |
| PaKA† | DINOv2R | **68.1** | **72.4** | **77.3** | **80.5** | **24.3** | **27.3** | **34.7** | **39.9** |

**Visual In-Context Learning.** We evaluate our method using the visual in-context reasoning benchmark (Balazevic et al., 2023b) which is inspired by natural language processing (NLP). This benchmark assesses a vision encoder's scene understanding capabilities directly from its learned representations, without relying on decoders or parameter tuning. Essentially, the benchmark performs patch-level nearest neighbor retrieval. Dense representations extracted from validation set images serve as queries, and the keys for retrieval are constructed from training images and stored in a memory bank. Label for a given query patch is subsequently predicted by leveraging the labels of its nearest neighbors retrieved from this memory bank.

We applied our post-training method to the pretrained models DINOv2R, and PaKA outperformed all competing methods in Table 1. On average, PaKA improved the performance of DINOv2R by 4.8% across all sampling fractions on PascalVOC. The high performance of PaKA in visual in-context learning suggests that our method excels at extracting semantically more similar features from the memory bank, which is a collection of features serving as a surrogate for the vast feature space.

**Overclustering.** The quality of the dense representations is further assessed using an overclustering task that requires minimal additional supervision similar to visual in-context learning. Following the previous work (Ziegler & Asano, 2022), we apply $K$-means clustering via Faiss (Johnson et al., 2019)

Table 2: **Overclustering-based evaluations.** Models are assessed by applying $K$-means clustering with a range of granularities $K$, which is ground-truth, and overclustering values of 300 and 500.

| Method | Pretrained | Pascal VOC | | | ADE20K | | |
|--------|------------|--------------|---------------|---------------|--------------|---------------|---------------|
| | | $K = \text{GT}$ | $K = 300$ | $K = 500$ | $K = \text{GT}$ | $K = 300$ | $K = 500$ |
| DINO | ✗ | 4.3 | 13.9 | 17.3 | 4.2 | 5.3 | 5.9 |
| iBOT | ✗ | 4.4 | 23.8 | 31.1 | 5.3 | 7.1 | 8.4 |
| CrOC | ✗ | 3.4 | 16.4 | 20.0 | 1.9 | 3.1 | 4.1 |
| CrIBo | ✗ | 18.3 | 51.3 | 54.5 | 7.3 | 9.6 | 11.9 |
| DINOv2R | ✗ | 12.2 | 46.7 | 49.5 | 7.5 | 9.8 | 11.5 |
| NeCo[†] | DINOv2R | 20.7 | 68.6 | 71.3 | 11.5 | 16.5 | 19.9 |
| PaKA[†] | DINOv2R | **21.3** | **74.5** | **76.5** | **12.1** | **17.4** | **21.4** |

Figure 4: **Visualization of Overclustering.** Results of Overclustering for DINOv2R, NeCo, and PaKA on Pascal VOC. Colored overlays represent matched semantic clusters derived via K-means.

to all dense features from the backbone, explicitly discarding the projection head. The generated clusters are greedily matched to ground-truth classes at the pixel level, followed by a Hungarian matching (Kuhn, 1955) to ensure permutation-invariant (Ji et al., 2019) evaluation. Performance is quantified using mIoU and assessed at various granularities, with $K$ set to the number of ground-truth objects as well as overclustering values of 300 and 500.

As shown in Table 2 and Figure 4, PaKA demonstrates its ability to learn expressive dense representations. In particular, for PascalVOC with $K = 500$, the post-trained NeCo model improves upon the DINOv2R baseline by 21.8 %, whereas PaKA attains a 27 % gain. This represents a 5.2 % absolute advantage over NeCo, which is an another post-training method. Such a margin suggests that PaKA learns features that are well-distributed in cluster-level within the feature space.

Table 3: **Linear segmentation performance.** Linear segmentation performance (mIoU) using heads trained on frozen spatial features from various extractors, evaluated on four datasets.

| Method | Pretrained | Pascal VOC | COCO-Things | COCO-Stuff | ADE20K |
|--------|------------|------------|-------------|------------|--------|
| DINO | ✗ | 50.2 | 43.9 | 45.9 | 17.5 |
| TimeT | DINO | 66.3 | 58.2 | 48.7 | 20.7 |
| iBOT | ✗ | 66.1 | 58.9 | 51.5 | 21.8 |
| CrOC | ✗ | 67.4 | 64.3 | 51.2 | 23.1 |
| CrIBo | ✗ | 71.6 | 64.3 | 49.1 | 22.7 |
| DINOv2R | ✗ | 74.2 | 75.3 | 56.0 | 35.0 |
| NeCo[†] | DINOv2R | 81.4 | 81.1 | 61.4 | 39.9 |
| PaKA[†] | DINOv2R | **82.2** | **82.5** | **62.6** | **40.9** |

**Linear Semantic Segmentation.** To assess the generalization capabilities of the learned representations, we perform linear semantic segmentation. This involves keeping the backbone weights frozen and attaching a linear classification head. The head projects dense features from the backbone to class logits, which are resized to the input resolution through bilinear interpolation to match the target masks. We compute pixel-level cross-entropy loss for supervision and report mIoU performance on four standard segmentation benchmarks. This approach directly evaluates the discriminative power of the static pretrained features, offering insights into their quality without task-specific adaptation of the backbone.

In Table 3, PaKA outperforms our base model DINOv2R by 5.9% to 8% across all four benchmarks and consistently outperforms NeCo. This demonstrates that PaKA still delivers the bigger boost, underscoring its superior ability to learn highly transferable, discriminative features without any task-specific fine-tuning of the backbone.

Table 4: **Semantic Correspondence.** Semantic correspondence matching comparison on SPair-71k (Min et al., 2019). We report both PCK per image ($PCK_{image}$) and PCK per point ($PCK_{point}$). PaKA boosts DINOv2R by +2.7 (+6.9%) in $PCK_{image}$ and +2.4 (+5.4%) in $PCK_{point}$.

| Metric | Method | Pretrained | SPair-71K Category | | | | | | | | | | | | | | | | | |
|--------|--------|-----------|------|------|------|------|--------|------|------|------|-------|------|------|-------|-------|--------|-------|-------|-------|------|
| | | | Aero | Bike | Bird | Boat | Bottle | Bus | Car | Cat | Chair | Cow | Dog | Horse | Motor | Person | Plant | Sheep | Train | TV | All |
| $PCK_{image}$ | DINOv2R | ✗ | 44.8 | 32.4 | 73.5 | 19.5 | 42.4 | 28.0 | 27.4 | 69.5 | 28.7 | 55.2 | 46.3 | 41.9 | 27.9 | 46.4 | 26.6 | 34.7 | 33.4 | 31.7 | 39.2 |
| | NeCo | DINOv2R | 42.7 | 33.8 | 76.6 | 21.3 | 44.0 | 28.9 | 28.1 | 65.3 | 31.0 | 55.5 | 47.3 | 44.3 | 29.9 | 53.8 | 37.1 | 40.6 | 38.5 | 39.2 | 42.0 |
| | PaKA | DINOv2R | 45.1 | 33.6 | 72.8 | 20.7 | 43.7 | 30.3 | 29.5 | 66.4 | 30.7 | 54.8 | 45.4 | 46.5 | 31.9 | 51.5 | 36.9 | 38.1 | 42.7 | 36.4 | 41.9 |
| $PCK_{point}$ | DINOv2R | ✗ | 47.4 | 34.8 | 75.3 | 22.9 | 44.9 | 38.5 | 39.8 | 69.4 | 31.9 | 61.3 | 49.5 | 45.5 | 30.8 | 50.5 | 29.6 | 42.7 | 36.3 | 35.5 | 44.4 |
| | NeCo | DINOv2R | 44.6 | 35.7 | 77.5 | 25.5 | 45.6 | 39.4 | 39.9 | 64.8 | 34.2 | 59.6 | 50.2 | 46.1 | 32.0 | 57.8 | 37.4 | 46.1 | 41.3 | 42.8 | 46.4 |
| | PaKA | DINOv2R | 47.3 | 35.6 | 74.5 | 25.4 | 45.6 | 41.3 | 41.8 | 66.1 | 33.8 | 59.6 | 48.4 | 48.3 | 34.2 | 55.6 | 37.6 | 45.0 | 45.5 | 39.7 | 46.8 |

(a) t-SNE of DINOv2     (b) t-SNE of NeCo     (c) t-SNE of PaKA

Figure 5: **Inter-Class Separation on Pascal VOC.** t-SNE visualization of dense patch features using 1,000 samples per category from Pascal VOC, showing that PaKA achieves clearer separation among semantically similar categories (e.g., *dog, horse*) compared to DINOv2 and NeCo, while NeCo exhibits overlap between these classes (highlighted in red circle in Figure 5b).

### 5.3 ENHANCED SEMANTIC DISCRIMINABILITY AND MATCHING

Ideally, dense representations should exhibit high semantic discriminability to support fine-grained downstream tasks. To verify whether structural alignment via CKA effectively enhances this property, we analyze the topology of the learned feature space. As visualized in Figure 5, the dense representations learned by PaKA form distinct and well-separated clusters compared to the baseline DINOv2. Notably, PaKA demonstrates superior inter-class separation even among semantically similar categories (e.g., *cat, dog,* and *horse*), whereas DINOv2 and NeCo exhibit blurred boundaries. This qualitative evidence suggests that PaKA's kernel-based objective captures richer structural dependencies, encouraging the model to disentangle subtle semantic differences that are often overlooked by standard alignment methods.

To quantitatively validate that this improved separation translates to robust matching capabilities, we evaluated the learned representations on semantic correspondence (Tang et al., 2023; Zhang et al., 2023; 2024). Unlike standard keypoint matching (DeTone et al., 2018; Sun et al., 2021) which targets identical scenes or objects, semantic correspondence requires identifying semantically corresponding points across different instances of the same category, making it a challenging benchmark for dense features. Evaluating on the SPair-71k (Min et al., 2019) benchmark, we observe in Table 4 that PaKA significantly outperforms the DINOv2 baseline, registering gains of +2.7 (+6.9%) in PCK_image and +2.4 (+5.4%) in PCK_point. These results underscore the effectiveness of PaKA in learning dense representations that are structurally aligned across diverse samples of the same category.

### 5.4 COMPUTATIONAL AND MEMORY EFFICIENCY ANALYSIS

We evaluated the resource efficiency of our proposed PaKA framework against NeCo, as shown in Table 5. For computational efficiency, we measured the total training time required for 25 epochs on a single NVIDIA H100 GPU. For memory efficiency, we assessed the maximum batch size that

Table 5: **Resource efficiency and performance on Pascal VOC.** Comparison of GPU hours, memory cost, and mIoU scores for NeCo and PaKA on post-training DINOv2R with a NVIDIA H100 GPU.

| Method | Pretrained | GPU Hours | Memory Cost | K=GT | K=300 | K=500 | Linear |
|--------|-----------|-----------|-------------|------|-------|-------|--------|
| DINOv2R | ✗ | - | - | 12.2 | 46.7 | 49.5 | 74.2 |
| NeCo | DINOv2R | 22 h 24 min | 1.90 GB per sample | 20.7 $^{\uparrow 8.5}$ | 68.6 $^{\uparrow 21.9}$ | 71.3 $^{\uparrow 21.8}$ | 81.4 $^{\uparrow 7.2}$ |
| PaKA | DINOv2R | 14 h 4 min $^{\downarrow 37\%}$ | 1.45 GB per sample $^{\downarrow 24\%}$ | 21.3 $^{\uparrow 9.1}$ | 74.5 $^{\uparrow 27.8}$ | 76.5 $^{\uparrow 27.0}$ | 82.2 $^{\uparrow 8.0}$ |

Table 6: **Ablation studies for PaKA.** To evaluate the ablated models, we employ overclustering ($K = GT$ and $K = 500$) and linear segmentation (Linear) on Pascal VOC 2012.

| (a) Kernel Alignment Metrics | | | |
|---|---|---|---|
| **Metric** | $K = GT$ | $K = 500$ | Linear |
| MMD | 19.6 | 65.3 | 76.8 |
| HSIC | 17.5 | 63.0 | 78.5 |
| Gram | 18.0 | 76.0 | **82.2** |
| CKA | **21.3** | **76.5** | **82.2** |

| (b) Augmentation Components | | | | |
|---|---|---|---|---|
| Max. | Clean. | $K = GT$ | $K = 500$ | Linear |
| | | 18.1 | 75.0 | 82.1 |
| ✓ | | 18.7 | 75.8 | 82.0 |
| | ✓ | 19.9 | 75.5 | 81.8 |
| ✓ | ✓ | **21.3** | **76.5** | **82.2** |

| (c) Loss-Augmentation Synergy | | | | |
|---|---|---|---|---|
| **Method** | Aug. | $K = GT$ | $K = 500$ | Linear |
| NeCo | | 20.7 | 71.3 | 81.4 |
| NeCo | ✓ | 17.8 | 73.1 | 81.5 |
| PaKA | | 18.1 | 75.0 | 82.1 |
| PaKA | ✓ | **21.3** | **76.5** | **82.2** |

could fit within a fixed VRAM capacity and used this to approximate memory usage. The results show that PaKA achieves strong dense representation performance within only 14 hours of training on a single GPU, which is 37% faster and 24% more memory-efficient compared to NeCo.

## 5.5 ABLATION STUDY

To validate the design choices of our proposed Patch-level Kernel Alignment (PaKA) framework and to understand the individual contributions of its key components, we conduct a series of ablation studies. All studies are conducted on the Pascal VOC 2012, reporting performance in overclustering tasks ($K = GT$, $K = 500$) and linear semantic segmentation tasks.

**Impact of Kernel Alignment Metric.** We ablated the choice of kernel alignment metric, comparing our Centered Kernel Alignment (CKA) against Maximum Mean Discrepancy (MMD), Hilbert-Schmidt Independence Criterion (HSIC), and Gram matrix (Gram) used in Gram anchoring objective (Siméoni et al., 2025). Table 6(a) reveals that CKA consistently achieves the highest scores across all datasets. This superior performance underscores CKA's greater efficacy in aligning dense feature distributions between teacher and student models, validating its use in PaKA.

**Efficacy of Proposed Augmentation Strategies.** Our ablation study assesses two strategies, maximizing global-local intersection (Max.) and a clean teacher (Clean.), against a baseline of standard augmentations (Ziegler & Asano, 2022) combined with the CKA loss. While Table 6(b) shows both provide individual gains, the maximizing intersection strategy's contribution is comparatively modest. This may be because standard augmentations already provide a mean view overlap of 75.5% (measured across 300 COCO samples), whereas our strategy enforces an even stricter $\geq 90\%$ overlap to maximize shared information. Importantly, combining the two augmentation strategies delivers the highest overall performance, demonstrating their synergistic benefits.

**Effectiveness of Our Augmentation Strategy.** We conducted a cross-combination study to evaluate the interplay between alignment losses (PaKA and NeCo (Pariza et al., 2025)), and augmentation strategies (our augmentation and standard augmentation (Ziegler & Asano, 2022)). Table 6(c) indicates that our proposed augmentation improves NeCo's sorting loss by +1.3% in overclustering K=500, which demands more fine-grained features. Notably, even when paired with standard augmentation, PaKA loss with standard augmentation already outperforms this enhanced NeCo performance. The best results across all datasets are achieved by combining PaKA loss with our proposed augmentation, highlighting their combined strength.

## 6 CONCLUSION

We presented a novel framework that significantly advances dense self-supervised learning by overcoming key limitations of existing methods. Our core contribution, PaKA, introduces kernel alignment for effective teacher-student dense feature structure transfer, utilizing Centered Kernel Alignment for efficient, assumption-free distributional matching. PaKA notably circumvents the need for auxiliary components such as iterative clustering, memory banks, or sorting algorithms, thereby reducing the training complexity and hyperparameter sensitivity frequently associated with these prior mechanisms. This streamlined approach, combined with our augmentations, establishes a robust framework achieving state-of-the-art performance on tasks requiring detailed spatial understanding, paving the way for more accessible and scalable dense SSL.

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

# Patch-level Kernel Alignment
# for Dense Self-Supervised Learning

## Supplementary Material

## A EXPERIMENTAL SETUP

### A.1 DENSE SELF-SUPERVISED LEARNING

**Dataset.** The pretraining datasets employed in this study include COCO. The COCO dataset, specifically, comprises approximately 118,000 scene-centric images.

**Network Architecture.** For the architectural backbone, ViTs are utilized. More specifically, models are trained using the ViT-Small and ViT-Base architectures. Furthermore, adopting the methodologies Caron et al. (2021); Grill et al. (2020), a student-teacher learning paradigm is implemented. Within this framework, the teacher model parameters are updated via an EMA of the student model's parameters.

**Removing registers in Dinov2.** Our Primary experiments are based on DINOv2XR, removing the registers from the DINOv2R Darcet et al. (2023) to restore the original DINOv2 Caron et al. (2021) patch-based input structure. Unless noted otherwise, all ablation studies and hyperparameter searches are likewise carried out on the DINOv2XR backbone.

**Optimization.** Our model was trained on a single H100 GPU with 80GB memory vutilizing a mini-batch size of 55 per GPU. Optimization was performed using the AdamW Loshchilov & Hutter (2017) optimizer. The learning rate for the backbone was set to $5e-6$ with a weight decay of $0.04$. We used 20 worker processes for data loading. The exponential moving average that updates the teacher's weights follows a cosine schedule, increasing from an initial value of $0.99$ to a value of $1$. Our models, initialized from pretrained checkpoints, were generally post-trained for 25 CoCo epochs. However, for experiments specifically focused on data augmentation, we trained 10 epochs. Based on Caron et al. (2021), we employ a three-layer projection head with 2,048 hidden units per layer, Gaussian error linear unit activations Hendrycks & Gimpel (2016), and an output dimension of 256.

**Data Augmentation.** Dense SSL methods Ziegler & Asano (2022); Pariza et al. (2025) create different views using augmentations (e.g., multi-crop), provide the teacher one global view, while the student receives global and multiple local views. When employing the DINOv2 Oquab et al. (2023) framework, input images were processed into global crops of $518 \times 518$ pixels and local crops of $98 \times 98$ pixels. To enhance dense feature alignment for our COCO pretraining, we introduce crucial refinements to this process. Specifically, we set the teacher's augmentation strength to 0 to provide a noise-free target, while applying standard augmentations to the student's views with a strength of 1. Furthermore, to maximize mutual information and ensure local views are strongly anchored within the global context, we enforce a strict overlap requirement, ensuring that the intersection area between a teacher's global crop and any corresponding local crop must exceed $0.9$.

### A.2 EVALUATION SETUP

**Visual In-Context Learning.** Our evaluation methodology adheres to the visual in-context reasoning benchmark proposed by Balazevic et al. (2023a), designed to assess the scene understanding capabilities of vision encoders directly from their learned representations without necessitating decoders or subsequent parameter adjustments. The core of this benchmark involves a patch-level nearest neighbor retrieval process. Dense representations extracted from validation set images serve as queries. The keys for retrieval are constructed from training images, which are uniformly sub-sampled from the complete training set at fractions of 1 (full set), $1/8$, $1/64$, or $1/128$. Patches derived from these chosen training images are then encoded and compiled into a memory bank. The label for any given query patch is subsequently inferred by leveraging the labels of its nearest neighbors retrieved from this memory bank. We utilized the open implementation by Pariza et al. (2024), which is faithful to the original description by Balazevic et al. (2023a) and employs the

Table 7: **Depth Estimation.** We fine-tuned linear layers with frozen backbone to predict the depth of each pixel. The performance are reported on three metrics: RMSE, AbsRel, $\delta_1$. The models were trained on classification loss on monocular depth estimation benchmark NYUd.

| Method | Pretrained | Linear 1. | | | Linear 4. | | |
| --- | --- | --- | --- | --- | --- | --- | --- |
| | | RMSE ↓ | AbsRel ↓ | $\delta_1$ ↑ | RMSE ↓ | AbsRel ↓ | $\delta_1$ ↑ |
| DINO | ✗ | .996 | .386 | .464 | .587 | .180 | .722 |
| DINOv2 | ✗ | .461 | .146 | .821 | .435 | .137 | .843 |
| DINOv2R | ✗ | .452 | .142 | .826 | .446 | .139 | .832 |
| NeCo[†] | DINOv2R | .455 | .143 | .825 | .458 | .143 | .824 |
| PaKA[†] | DINOv2R | **.443** | **.139** | **.834** | **.426** | **.131** | **.850** |

ScaNN library Guo et al. (2020) for efficient neighbor searches. Consistent with the setup for the Balazevic et al. (2023a), we used a memory size of 10,240,000 and configured ScaNN to retrieve 30 nearest neighbors. Final performance is reported as mean Intersection over Union (mIoU) on subsets of Pascal VOC 2012 Everingham et al. and ADE20K Zhou et al. (2017) datasets.

**Overclustering.** To evaluate unsupervised segmentation quality via overclustering, we adopted the protocol from Leopart Ziegler & Asano (2022). The procedure commences with feature extraction: spatial tokens are gathered from the model's backbone using input images standardized to $448 \times 448$ crops. These tokens then undergo $K$-Means clustering, for which the faiss Johnson et al. (2019) implementation is employed. The crucial step of achieving permutation invariance, as emphasized by Ji et al. (2019), is realized by applying the Hungarian algorithm Kuhn (1955). This algorithm operates on cluster maps that are initially formed through a greedy matching process based on pixel-level precision. To ensure the computational feasibility of the Hungarian matching, the overclustering is performed on $100 \times 100$ downsampled masks. Final assessment metrics are reported as the average mean Intersection over Union (mIoU) from five runs with different random seeds, on the Pascal VOC 2012 Everingham et al., and ADE20K Zhou et al. (2017) datasets.

**Linear Semantic Segmentation.** The linear semantic segmentation evaluation followed the setup established in Leopart Ziegler & Asano (2022). The process of generating predictions began with 448x448 input images, which were fed into our backbone model to obtain spatial output features. These features were then resized using bilinear interpolation to align with the dimensions of the target segmentation masks. A linear classification head was subsequently applied to these processed features to yield the final segmentation predictions. The training of this linear head was driven by a cross-entropy loss, calculated between the predictions and the ground-truth masks. The optimization was performed using Stochastic Gradient Descent (SGD) with specific hyperparameters: a weight decay of 0.0001, momentum of 0.9, and a learning rate of 0.01. The linear head was fine-tuned over 20 epochs. We trained and evaluated these linear heads on Pascal VOC 2012 Everingham et al., COCO-Thing, COCO-Stuff Caesar et al. (2018), and ADE20K Zhou et al. (2017) datasets.

**End-to-End Finetuning with a linear head.** The end-to-end finetuning segmentation approach builds upon the previously detailed linear evaluation setup, retaining most of its core configurations. However, in this mode, we fine-tuned the entire model, thereby jointly optimizing the parameters of the feature-extracting backbone and the linear head. To facilitate this full network training, the backbone was finetuned using a learning rate of 0.0001, and the linear head was simultaneously trained with a learning rate of 0.01. The fine-tuning process was conducted for 20 epochs, with performance evaluated as mean Intersection over Union (mIoU) on Pascal VOC 2012 Everingham et al., COCO-Thing and COCO-Stuff Caesar et al. (2018), and ADE20K Zhou et al. (2017) datasets.

## B ADDITIONAL EXPERIMENTS

### B.1 DEPTH ESTIMATION.

For depth estimation, we evaluate our features on NYUd (Couprie et al., 2013) dataset, following the protocol of previous work Li et al. (2022b). To assess our patch-level features in pixel-wise evaluation, we measure depth estimation performance by finetuning linear heads while keeping the

Table 8: **Panoptic Segmentation on ADE20K.** Metrics include overall PQ, SQ, RQ and their Thing/Stuff variants: $PQ_{th}$, $SQ_{th}$, $RQ_{th}$ for *thing* classes, and $PQ_{st}$, $SQ_{st}$, $RQ_{st}$ for *stuff* classes.

| Method | Pretrained | PQ | SQ | RQ | $PQ_{th}$ | $SQ_{th}$ | $RQ_{th}$ | $PQ_{st}$ | $SQ_{st}$ | $RQ_{st}$ |
|---|---|---|---|---|---|---|---|---|---|---|
| DINOv2R | ✗ | 22.42 | **64.07** | 27.80 | 20.65 | **67.30** | 26.26 | 25.96 | 57.61 | 30.87 |
| NeCo | DINOv2R | 22.26 | 62.60 | 27.75 | 20.18 | 64.58 | 25.75 | 26.42 | 58.63 | 31.75 |
| PaKA | DINOv2R | **23.53** | 63.75 | **29.17** | **21.15** | 65.74 | **26.88** | **28.29** | **59.79** | **33.74** |

backbone frozen. The model is trained with either a single linear layer(Linear 1.) or a 4-layer linear head(Linear 4.) in Table 7, using a classification loss. We reported our performance on NYUd using various metric: root mean squared error (RMSE), accuracy under the threshold ($\delta_i < 1.25^i$, $i = 1$), and mean absolute relative error (AbsRel).

## B.2 PANOPTIC SEGMENTATION.

We further validated the effectiveness of PaKA by applying it to the challenging task of panoptic segmentation on ADE20K. For this experiment, we utilized the Mask2Former (Cheng et al., 2022), adapting our ViT-based backbone to output multi-scale features. Specifically, we extracted features from four intermediate layers corresponding to strides of 4, 8, 16, and 32, ensuring compatibility with the pixel decoder. We trained the model for 200k steps with a batch size of 4, setting the learning rate to $1 \times 10^{-4}$ for the decoder and $1 \times 10^{-5}$ for the backbone. As shown in Table 8, PaKA consistently outperforms the DINOv2R baseline and NeCo across key metrics, achieving the highest overall PQ of 23.53 and RQ of 29.17. The improvements are particularly pronounced in the stuff categories, where PaKA achieves a $PQ_{st}$ of 28.29 compared to 25.96 for DINOv2R.

Table 9: **End-to-End Fine-Tuning evalutaions with a linear head.** We fine-tuned various backbones with a linear head end-to-end, which is pretrained with different self-supervised learning methods. The mIoU scores are reported on four different datasets.

| Method | Pretrained | Pascal VOC | COCO-Things | COCO-Stuff | ADE20K |
|---|---|---|---|---|---|
| DINO | ✗ | 65.4 | 65.4 | 54.3 | 29.9 |
| iBOT | ✗ | 73.8 | 71.8 | 57.0 | 33.3 |
| CrIBo | ✗ | 75.7 | 73.1 | 55.6 | 33.4 |
| DINOv2R | ✗ | 81.5 | 82.2 | 61.9 | 42.5 |
| NeCo[†] | DINOv2R | 82.7 | 82.6 | 62.8 | 44.7 |
| PaKA[†] | DINOv2R | **83.0** | **83.1** | **63.3** | **45.1** |

## B.3 END-TO-END FINETUNING WITH A LINEAR HEAD.

To evaluate the transferability of pretrained features in an end-to-end setup, we finetune the entire network including the backbone and linear head. The model is trained using a pixel-wise cross-entropy loss, and final performance is reported using mIoU on Pascal VOC 2012, COCO-Things, COCO-Stuff 164K, and ADE20K.

After end-to-end fine-tuning, PaKA attains the highest mIoU scores on every benchmark compared to all dense self-supervised methods in Table 9. These consistent gains demonstrate that PaKA is a strong, task-agnostic initialization for dense prediction tasks. Even more surprisingly, the post-training method NeCo actually degrades the performance of its underlying pretrained model, while our approach constantly improves the preformance of DINOv2R.

## B.4 END-TO-END FINETUNING WITH A TRANSFORMER DECODER.

We further validated our method by integrating a Transformer-based decoding head to assess performance in a non-linear regime. Our implementation utilizes a decoder with 2 layers and 6 heads, inspired by the Segmenter (Strudel et al., 2021). This module operates by concatenating learnable class embeddings with image patch tokens, followed by joint processing through self-attention layers. The final segmentation masks are then generated by computing the scalar product between the refined class embeddings and the patch features. Table 10 summarizes the performance with

Table 10: **End-to-End Fine-Tuning evaluations using a Transformer decoder.** The model is fine-tuned end-to-end using a decoder with 2 transformer layers. PaKA demonstrates consistent improvements over the DINOv2 baseline in this non-linear evaluation setting.

| Method | Pretrained | Pascal VOC | COCO-Things | COCO-Stuff | ADE20K |
|--------|-----------|-----------|-------------|-----------|--------|
| DINOv2R | ✗ | 82.74 | **84.12** | 61.72 | 45.95 |
| NeCo | DINOv2R | 83.03 | 83.33 | 61.99 | 45.86 |
| PaKA | DINOv2R | **84.43** | 83.89 | **62.21** | **46.59** |

a Transformer decoder. PaKA attains the highest mIoU scores on Pascal VOC, COCO-Stuff, and ADE20K, demonstrating its effectiveness even with a non-linear evaluation protocol.

### B.5 VIT-B MODEL PERFORMANCE

To demonstrate the robustness and broader applicability of our Patch-level Kernel Alignment (PaKA) framework, this section evaluates its performance on the ViT-B architecture. Unlike the ViT-S experiments, the ViT-B experiments were conducted on four NVIDIA RTX 3090 GPUs (24GB VRAM each) due to resource constraints. For these experiments, the learning rate for the backbone was set to $1e-6$ with a weight decay of $0.04$. For a fair comparison, the NeCo baseline (Pariza et al., 2025) was also trained under the same hardware conditions. Although conducted in a more resource-constrained environment, we include these results to validate that the observed performance trends generalize to the larger base model.

Table 11: **Visual In-Context Learning benchmark.** Dense Nearest Neighbor retrieval performed on ADE20k and Pascal VOC datasets with varying training data proportions. We construct uniformly sampled training subsets, down-sampling the full set by ratios of 1/1, 1/8, 1/64, and 1/128.

| Method | Backbone | Pretrained | ADE20K | | | | Pascal VOC | | | |
|--------|----------|-----------|-------|------|------|------|-------|------|------|------|
| | | | 1/128 | 1/64 | 1/8 | 1/1 | 1/128 | 1/64 | 1/8 | 1/1 |
| DINO | ViT-S/16 | ✗ | 9.5 | 11.0 | 15.0 | 17.9 | 26.4 | 30.5 | 41.3 | 48.7 |
| SelfPatch | ViT-S/16 | ✗ | 10.0 | 10.9 | 14.7 | 17.7 | 28.4 | 32.6 | 43.2 | 50.8 |
| CrOC | ViT-S/16 | ✗ | 8.7 | 10.8 | 15.2 | 17.3 | 34.0 | 41.8 | 53.8 | 60.5 |
| Leopart | ViT-S/16 | DINO | 12.9 | 14.8 | 19.6 | 23.9 | 44.6 | 49.7 | 58.4 | 64.5 |
| CrIBo | ViT-S/16 | ✗ | 14.6 | 17.3 | 22.7 | 26.6 | 53.9 | 59.9 | 66.9 | 72.4 |
| DINOv2R | ViT-S/14 | ✗ | 23.7 | 27.1 | 33.9 | 39.5 | 60.1 | 65.7 | 74.5 | 78.8 |
| NeCo[†] | ViT-S/14 | DINOv2R | 23.9 | 27.2 | 34.3 | 39.8 | 65.5 | 69.0 | 75.1 | 78.8 |
| PaKA[†] | ViT-S/14 | DINOv2R | **24.3** | **27.3** | **34.7** | **39.9** | **68.1** | **72.4** | **77.3** | **80.5** |
| MAE | ViT-B/16 | ✗ | 10.0 | 11.3 | 15.4 | 18.6 | 3.5 | 4.1 | 5.6 | 7.0 |
| DINO | ViT-B/16 | ✗ | 11.5 | 13.5 | 18.2 | 21.5 | 33.1 | 37.7 | 49.8 | 57.3 |
| Leopart | ViT-B/16 | ✗ | 14.6 | 16.8 | 21.8 | 26.7 | 50.1 | 54.7 | 63.1 | 69.5 |
| Hummingbird | ViT-B/16 | ✗ | 11.7 | 15.1 | 22.3 | 29.6 | 50.5 | 57.2 | 64.3 | 71.8 |
| CrIBo | ViT-B/16 | ✗ | 15.9 | 18.4 | 24.4 | 28.4 | 55.9 | 61.8 | 69.2 | 74.2 |
| DINOv2R | ViT-B/14 | ✗ | 22.1 | 25.8 | 33.2 | 38.7 | 51.8 | 58.9 | 70.6 | 77.3 |
| NeCo[†] | ViT-B/14 | DINOv2R | 26.7 | 30.6 | 38.6 | 43.3 | 64.6 | 70.2 | 78.3 | 81.6 |
| PaKA[†] | ViT-B/14 | DINOv2R | **28.2** | **32.0** | **40.2** | **44.3** | **69.1** | **74.0** | **79.6** | **82.6** |

**Visual In-Context Learning on ViT-B.** The visual in-context learning results presented in Table 11 highlight the significant benefits of leveraging the larger ViT-B architecture with our PaKA framework, which not only yields overall performance gains over PaKA with ViT-S, but also markedly outperforms other methods. When compared to other methods employing the ViT-B backbone, PaKA consistently achieves the highest performance across all data regimes on both ADE20K and Pascal VOC datasets. This includes outperforming the prior post-training method NeCo (ViT-B), for example, by 4.5% (1/128) and 3.8% (1/64) on Pascal VOC. Such strong performance highlights that PaKA effectively learns a feature space well-structured for nearest-neighbor based scene understanding.

Table 12: **Linear segmentation performance.** Linear segmentation performance (mIoU) using heads trained on frozen spatial features from various extractors, evaluated on four datasets.

| Method | Backbone | Pretrained | Pascal VOC | COCO-Things | COCO-Stuff | ADE20K |
|---|---|---|---|---|---|---|
| DINO | ViT-S/16 | ✗ | 50.2 | 43.9 | 45.9 | 17.5 |
| TimeT | ViT-S/16 | DINO | 66.3 | 58.2 | 48.7 | 20.7 |
| iBOT | ViT-S/16 | ✗ | 66.1 | 58.9 | 51.5 | 21.8 |
| CrOC | ViT-S/16 | ✗ | 67.4 | 64.3 | 51.2 | 23.1 |
| CrIBo | ViT-S/16 | ✗ | 71.6 | 64.3 | 49.1 | 22.7 |
| DINOv2R | ViT-S/14 | ✗ | 74.2 | 75.3 | 56.0 | 35.0 |
| NeCo[†] | ViT-S/14 | DINOv2R | 81.4 | 81.1 | 61.4 | 39.9 |
| PaKA[†] | ViT-S/14 | DINOv2R | **82.2** | **82.5** | **62.6** | **40.9** |
| DINO | ViT-B/16 | ✗ | 62.7 | 55.8 | 51.2 | 23.6 |
| MAE | ViT-B/16 | ✗ | 32.9 | 38.0 | 38.6 | 5.8 |
| iBOT | ViT-B/16 | ✗ | 73.1 | 69.4 | 55.9 | 30.1 |
| CrIBo | ViT-B/16 | ✗ | 73.9 | 69.6 | 53.0 | 25.7 |
| DINOv2R | ViT-B/14 | ✗ | 80.2 | 84.8 | 59.3 | 43.0 |
| NeCo[†] | ViT-B/14 | DINOv2R | 84.3 | 85.8 | 64.0 | 45.2 |
| PaKA[†] | ViT-B/14 | DINOv2R | **84.4** | **85.9** | **64.4** | **46.1** |

**Linear Semantic Segmentation on ViT-B.** In linear semantic segmentation, leveraging the larger ViT-B/14 architecture with our PaKA framework leads to notable performance gains over PaKA with ViT-S/14. Crucially, PaKA trained on this ViT-B model achieves the highest mIoU scores across all benchmarks compared to other methods in Table 12. Using the ViT-B backbone, PaKA demonstrates a consistent advantage, achieving an average mIoU gain of 3.4 over DINOv2R across the benchmarks.

Table 13: **End-to-End FineTuning evaluations with a linear head.** We fine-tuned various backbones with a linear head end-to-end, which is pretrained with different self-supervised learning methods. The mIoU scores are reported on four different datasets.

| Method | Backbone | Pretrained | Pascal VOC | COCO-Things | COCO-Stuff | ADE20K |
|---|---|---|---|---|---|---|
| DINO | ViT-S/16 | ✗ | 65.4 | 65.4 | 54.3 | 29.9 |
| iBOT | ViT-S/16 | ✗ | 73.8 | 71.8 | 57.0 | 33.3 |
| CrIBo | ViT-S/16 | ✗ | 75.7 | 73.1 | 55.6 | 33.4 |
| DINOv2R | ViT-S/14 | ✗ | 81.5 | 82.2 | 61.9 | 42.5 |
| NeCo[†] | ViT-S/14 | DINOv2R | 82.7 | 82.6 | 62.8 | 44.7 |
| PaKA[†] | ViT-S/14 | DINOv2R | **83.0** | **83.1** | **63.3** | **45.1** |
| DINOv2R | ViT-B/14 | ✗ | 83.9 | 86.0 | 63.1 | 47.2 |
| NeCo[†] | ViT-B/14 | DINOv2R | 85.0 | **86.4** | 64.3 | 47.4 |
| PaKA[†] | ViT-B/14 | DINOv2R | **85.1** | **86.4** | **64.8** | **48.4** |

**End-to-End Finetuning Segmentation on ViT-B.** When the entire network is finetuned for semantic segmentation, PaKA's ViT-B initialization proves superior. Crucially, PaKA trained on this ViT-B model achieves the highest mIoU scores across all benchmarks compared to other methods. Following Table 13, this advantage is particularly pronounced on the challenging COCO-Stuff and ADE20K datasets. On COCO-Stuff, PaKA reaches 64.8%, and on ADE20K, it achieves 48.4%, setting new state-of-the-art results in both cases and clearly outperforming NeCo. This demonstrated that PaKA generalize and learn robust features applicable to diverse and complex semantic segmentation tasks.

B.6 DIVERSE PRETRAINED BACKBONES

We applied PaKA into different backbones initialized with different pretraining methods. Experiments were conducted on four NVIDIA RTX 3090 GPUs (24GB VRAM each), with the learning rate $1e-6$. The linear segmentation results, presented in Table 14, show that PaKA consistently and significantly

improves performance across all four benchmark datasets. Notably, when PaKA is applied to an image-level pretrained model, DINO Caron et al. (2021), it yields substantial gains, such as +15.0 mIoU on Pascal VOC and +17.4 mIoU on COCO-Things. Furthermore, PaKA also enhances models already pretrained with dense SSL objectives. For instance, it improves iBOT Zhou et al. (2022) by +4.0 mIoU on Pascal VOC and +7.9 mIoU on ADE20K, and improves CrIBo Lebailly et al. (2024) by +5.1 mIoU on COCO-Stuff and +6.2 mIoU on ADE20K. This consistent uplift underscores PaKA's ability to effectively refine and adapt features for dense prediction tasks.

Table 14: **PaKA's Enhancement of Diverse Pretrained Models for Linear Segmentation.** This table details the linear segmentation performance achieved by applying PaKA post-training to various pretrained models initialized with DINO Caron et al. (2021), iBOT Zhou et al. (2022), and CrIBo Lebailly et al. (2024), showcasing its adaptability.

| Method | Backbone | Pretrained | Pascal VOC | COCO-Things | COCO-Stuff | ADE20K |
|---|---|---|---|---|---|---|
| DINO Caron et al. (2021) | ViT-S/16 | ✗ | 50.2 | 43.9 | 45.9 | 17.5 |
| + PaKA | ViT-S/16 | DINO | $65.2^{\uparrow 15.0}$ | $61.3^{\uparrow 17.4}$ | $54.0^{\uparrow 8.1}$ | $27.2^{\uparrow 9.7}$ |
| iBOT Zhou et al. (2022) | ViT-S/16 | ✗ | 66.1 | 58.9 | 51.5 | 21.8 |
| + PaKA | ViT-S/16 | iBOT | $70.1^{\uparrow 4.0}$ | $66.4^{\uparrow 7.5}$ | $57.2^{\uparrow 5.7}$ | $29.7^{\uparrow 7.9}$ |
| CrIBo Lebailly et al. (2024) | ViT-S/16 | ✗ | 71.6 | 64.3 | 49.1 | 22.7 |
| + PaKA | ViT-S/16 | CrIBo | $73.4^{\uparrow 1.8}$ | $68.1^{\uparrow 3.8}$ | $54.2^{\uparrow 5.1}$ | $28.9^{\uparrow 6.2}$ |

Table 15: **Applying PaKA to Vision-Language Model.** We conducted overclustering task with a range of granularities K: the ground-truth number of classes (GT), 300, and 500.

| Method | Pretrained | K=GT | K=300 | K=500 |
|---|---|---|---|---|
| CLIP | - | 15.0 | 49.3 | 50.9 |
| NeCo | CLIP | 14.2 | 52.2 | 55.7 |
| PaKA | CLIP | **16.6** | **54.3** | **56.9** |

Table 16: **Zero-Shot Segmentation in CLIP.** We evaluate post-trained CLIP vision encoders with the original CLIP text encoder under a zero-shot segmentation setup across COCO-Stuff, ADE20K, and Pascal-Context.

| Method | Pretrained | COCO-Stuff | | ADE20K | | Pascal-Context | |
|---|---|---|---|---|---|---|---|
| | | mIoU | mAcc | mIoU | mAcc | mIoU | mAcc |
| CLIP | - | **14.5** | **32.8** | **9.1** | **25.1** | 21.9 | **44.0** |
| NeCo | CLIP | 11.7 | 29.1 | 7.1 | 22.0 | 17.8 | 38.9 |
| PaKA | CLIP | 14.1 | 31.8 | 8.4 | 23.3 | **21.6** | 43.6 |

## B.7 VISION-LANGUAGE MODEL

We further applied PaKA to a Vision-Language Model, CLIP Radford et al. (2021), by post-training its vision encoder. Improving Vision Understanding in CLIP: As shown in Table 15, PaKA significantly improves the overclustering performance of CLIP, achieving substantial gains across all granularities (K=GT, K=300, K=500), compared to another post-training method, NeCo. This confirms that our method generalizes well to non-DINO backbones and is not tied to a specific pretraining architecture.

To further assess whether PaKA maintains cross-modal consistency, we evaluated the post-trained vision encoder in a zero-shot segmentation setting with the pre-trained CLIP text encoder in Table 16. Although PaKA does not explicitly optimize cross-modal alignment, only a marginal drop is observed,

Table 17: **Applying PaKA to other DINO Backbones.** We conduct overclustering with $K = $ GT, 300, and 500. PaKA consistently improves performance across different DINO backbones.

| Method | Pretrained | Pascal VOC | | | ADE20K | | |
|--------|-----------|------|------|------|------|------|------|
| | | K=GT | K=300 | K=500 | K=GT | K=300 | K=500 |
| DINOv2 | - | 13.8 | 47.6 | 54.2 | 7.9 | 10.2 | 13.0 |
| PaKA ↑ | DINOv2 | 12.4 | 55.5 | 59.8 | 10.0 | 13.8 | 17.0 |
| DINOv3 | - | 16.1 | 55.2 | 60.4 | 9.9 | 13.3 | 16.6 |
| NeCo ↓ | DINOv3 | 12.7 | 50.0 | 55.0 | 8.8 | 11.5 | 14.1 |
| PaKA ↑ | DINOv3 | 16.7 | 58.9 | 62.8 | 10.2 | 13.7 | 16.7 |

and the degradation is significantly smaller than that of NeCo. This shows that PaKA preserves text–vision alignment much more effectively while enhancing dense vision understanding.

## B.8 DINOV3

We further evaluate the effectiveness of our method on DINOv3 Siméoni et al. (2025) that stabilizes long training with Gram anchoring. As shown in Table 17, PaKA consistently enhances the dense understanding capability of DINOv3, despite its already strong baseline. Notably, this improvement is achieved with extremely low training costs, requiring only 15 hours of training on a single H100 GPU, which is a fraction of the cost compared to pretraining.

Table 18: **Impact of Batch Size on Performance.** We conduct overclustering on the Pascal VOC dataset with cluster granularities $K \in \{GT, 300, 500\}$.

| | In-context | | | | Overclustering | | |
|--------|------|------|------|------|------|------|------|
| Batch size | 1/128 | 1/64 | 1/8 | 1/1 | K=GT | K=300 | K=500 |
| 8 | 60.5 | 65.5 | 72.8 | 77.4 | 14.2 | 62.2 | 65.5 |
| 16 | 65.9 | 70.6 | 76.1 | 79.2 | 20.7 | 70.4 | 73.0 |
| 32 | 66.4 | 70.8 | 76.6 | 79.4 | 21.3 | 72.8 | 75.0 |
| 64 | 67.4 | 71.6 | 77.1 | 80.3 | 20.2 | 74.0 | 76.4 |

## B.9 BATCH SIZE ABLATION

We conducted a batch size ablation study. As shown in the Table 18, we observe a consistent improvement in downstream performance as the batch size increases. We expect that the performance could further improve with even larger batch sizes, given sufficient computational resources.

Table 19: **Classification on ImageNet.** We evaluate global representation quality on ImageNet using k-NN with $k \in \{10, 50, 100\}$ and linear probing using [CLS] token features.

| Method | Metric | DINOv2 | NeCo | PaKA |
|--------|--------|--------|------|------|
| **k=10** | Top-1 | **79.02** | 74.79 | 75.79 |
| | Top-5 | **91.62** | 89.12 | 89.70 |
| **k=50** | Top-1 | **78.57** | 74.13 | 75.18 |
| | Top-5 | **93.96** | 91.76 | 92.46 |
| **k=100** | Top-1 | **77.91** | 73.35 | 74.49 |
| | Top-5 | **94.29** | 92.06 | 92.74 |
| **Linear** | Top-1 | **79.26** | 76.69 | 77.89 |
| | Top-5 | **94.77** | 93.52 | 94.16 |

### B.10 GLOBAL UNDERSTANDING AND CLASSIFICATION ABILITY

We conducted a k-NN classification experiment on ImageNet using the [CLS] token to evaluate global understanding, comparing PaKA to the DINOv2 baseline in Table 19. The results indicate that PaKA's performance on this global task is slightly lower than that of the original DINOv2. We interpret this as a direct trade-off of our objective: while DINOv2 distills global information primarily into the specialized [CLS] token, PaKA's alignment objective encourages this context to be distributed across the individual patch tokens, thereby enhancing them for dense tasks at the expense of the [CLS] token's specialization.

## C QUALITATIVE EVALUATIONS

### C.1 DENSE NEAREST NEIGHBOR RETRIEVAL

Figure 6, derived from our visual in-context learning evaluation on Pascal VOC, illustrates PaKA's superior semantic understanding in dense nearest patch retrieval compared to DINOv2R. For each query patch, the top five nearest neighbors are retrieved from the dataset. A striking example is when a query patch depicting an airplane tail is presented: DINOv2R retrieves largely irrelevant patches, such as a bicycle wheel and horses. In contrast, PaKA for the same query retrieves highly relevant patches of airplane tails, reflecting its deeper understanding of patch-level semantics and object structure.

### C.2 OVERCLUSTERING

To qualitatively assess the semantic grouping capabilities inherent in different encoders, we generate visualizations directly from their features without any decoders or finetuning. The process begins by extracting dense patch-level features from validation images using the frozen model backbone. These features then undergo dimensionality reduction via Principal Component Analysis before being grouped using K-Means overclustering. This yields a low resolution map where each patch is assigned a raw cluster ID. To facilitate semantic interpretation, these raw clusters are matched to ground-truth semantic classes using a many-to-one mapping that maximizes IoU. The resulting visualizations in Figure 7 visually confirm that PaKA yields significantly cleaner clusters that better adhere to object boundaries and capture fine-grained details, outperforming both DINOv2R and NeCo in this qualitative assessment.

## D DATASET DETAILS

**Pascal VOC 2012 Everingham et al.**   The Pascal VOC 2012 presents natural images primarily centered on everyday objects, providing clear examples for object recognition and segmentation. This dataset (trainaug split) provides 10,582 training and 1,449 validation images, covering 21 semantic classes including background.

**COCO Lin et al. (2014)**   COCO images depict complex everyday scenes, often featuring multiple objects in their natural context with varying scales and significant occlusions. This dataset provides 118,000 training images and 5,000 validation images, annotated across 80 distinct object categories. In our work, we utilize the *train2017* and *val2017* splits from the COCO dataset.

**COCO-Stuff 164K Caesar et al. (2018)**   The COCO-Stuff 164K dataset features complex, everyday scenes densely populated with multiple distinct objects ("thing") and large, amorphous regions ("stuff"). It includes detailed annotations for 91 "stuff" categories and 80 "thing" categories. Following Pariza et al. (2025), , we consolidate these into a reduced set of 15 "stuff" and 12 "thing" classes for our evaluations.

**ADE20K Zhou et al. (2017)**   The ADE20K dataset is recognized for its challenging and diverse scenes, featuring highly detailed annotations. It includes 20,210 images in its training set and 2,000 images for validation. The dataset encompasses 150 unique semantic categories, covering a wide array of "stuff" (e.g., sky, grass) and "objects" (e.g., person, car). The "others" label is excluded from our evaluations.

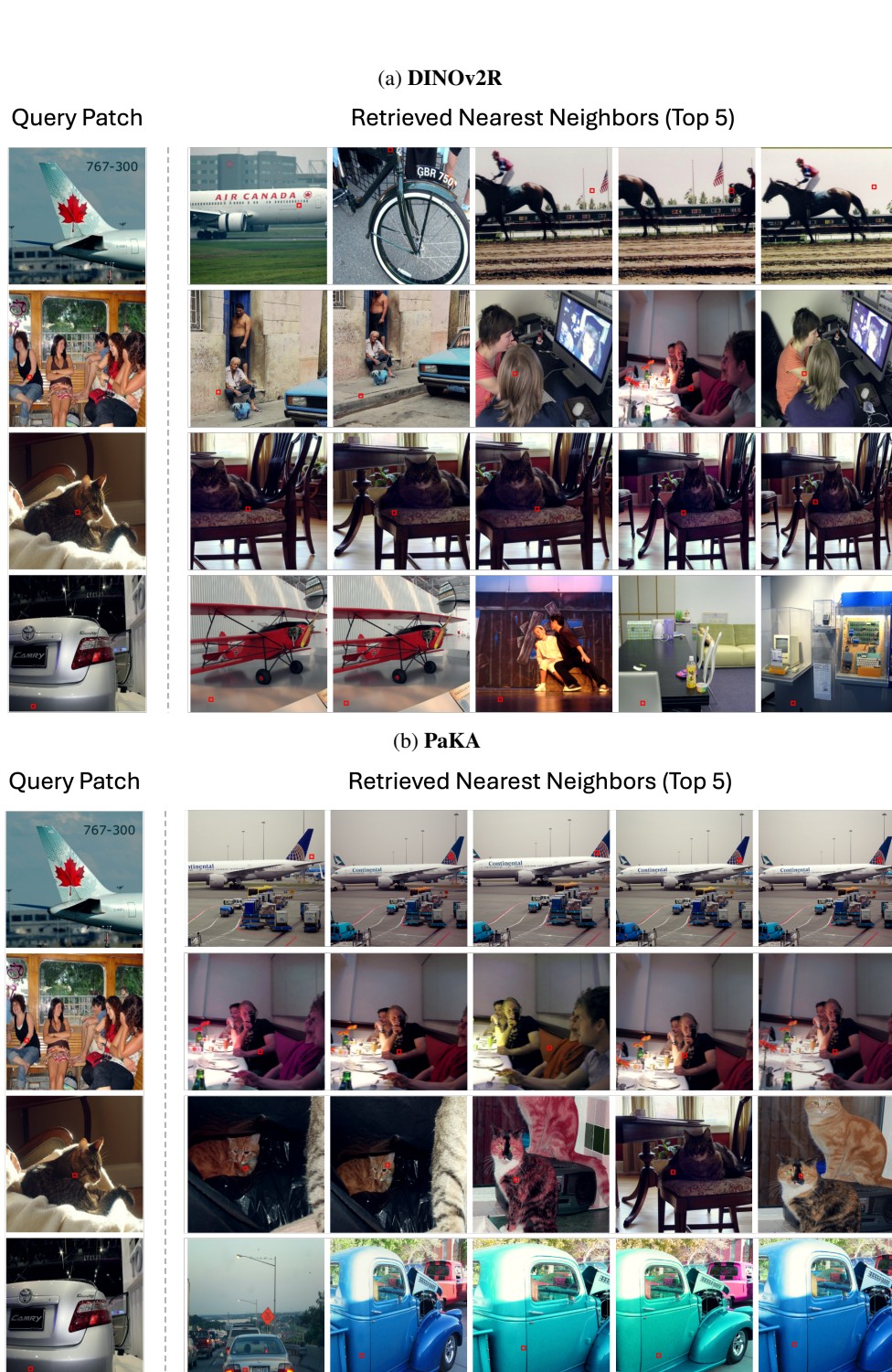

Figure 6: **Visualization of Vision In-Context Learning.** This figure contrasts the top five nearest neighbors retrieved by PaKA versus DINOv2R on Pascal VOC. PaKA consistently finds more semantically relevant and precise patches, including specific object parts.

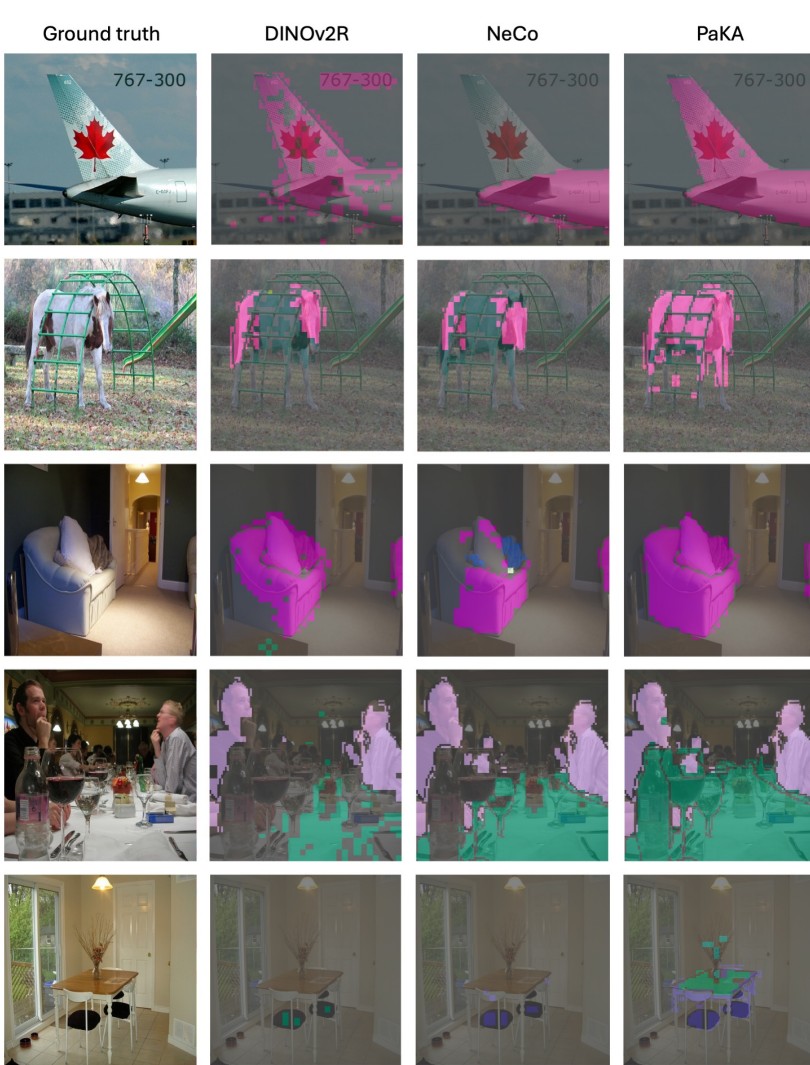

Figure 7: **Visualization of Overclustering.** Results of Overclustering for DINOv2R, NeCo, and PaKA on Pascal VOC. Colored overlays represent matched semantic clusters derived via K-means, visually demonstrating PaKA's superior ability to capture cleaner, object-level groupings compared to DINOv2R and NeCo.

