# OpenReview forum: "Patch-Level Kernel Alignment for Dense Self-Supervised Learning"
_ICLR.cc/2026/Conference — Submitted to ICLR 2026_

### Official Review · Reviewer_CCN1 · 2025-10-28

**Soundness:** 1
**Presentation:** 2
**Contribution:** 2
**Rating:** 2
**Confidence:** 3

**Summary:**

This paper investigates SSL loss for dense-prediction tasks. The paper proposes a lightweight post-training phase with a new SSL loss to improve the dense-feature quality of a DINOv2-pretrained backbone. The paper's main contributions are:
1) A new PaKA loss that adopts Centered Kernel Alignment (CKA) to compare patch distributions between student and teacher views, after alignment of the local patches through an ROI process.
2) A new augmentation strategy that maximizes overlap between student and teacher views and reduces the strength of teacher data augmentation.

The authors validate their approach by post-training a DINOv2R baseline and testing it on VOC and ADE20K using both visual in-context learning and linear probing. Their approach shows improvement over DINOv2R or NeCo baselines. The authors also perform an ablation to show the impact of the different contributions.

**Strengths:**

- Paper introduces a lightweight post-training to performance of DINOv2 backbone
- Paper performs careful ablation to study the impact of the different components (CKA loss, data-augmentation)

**Weaknesses:**

My main issue with the paper is that the empirical results do not seem to fully support the claim:
- It seems that the reported numbers are quite low for the baselines. For instance, Table 3 in [1] reports that DINOv2 obtains a score of 83.1 on PascalVOC and 49.5 on ADE20K. In contrast, DINO2R is getting around 74.2 and 35.0 on VOC/ADE20K in Table 2. Are you using a different base model, and would the proposed approach transfer to a stronger base model?
- CKA is proposed as one of the main contributions in the paper but does not lead to a significant improvement in the linear probing protocol in Table 5a.
- The authors do not control for post-training on a different data distribution (COCO). What would be the performance of the DINO baseline, post-trained on COCO using the regular DINO loss and gram anchoring?
- Other supervised baselines could be included, such as PESpatial or AM-RADIOv2.5, and the evaluation could go beyond the linear probing protocol.

[1]: DINOv3, Siméoni et al., 2025.

**Questions:**

My main question is related to the first weakness, why do we see a gap in term of performance on dense linear probing compare to result reported to the litterature and would the gain obtain by the approach transfer to stronger backbone ?

---

> ### Author Response · Authors · 2025-11-26
>
> We appreciate your thoughtful concerns, and we are glad to have the opportunity to address them. Our responses to your comments are as follows.
>
> &nbsp;
> > **[Concerns About Reported Baseline Performance]** _It seems that the reported numbers are quite low for the baselines. For instance, Table 3 in [1] reports that DINOv2 obtains a score of 83.1 on PascalVOC and 49.5 on ADE20K. In contrast, DINO2R is getting around 74.2 and 35.0 on VOC/ADE20K in Table 2. Are you using a different base model, and would the proposed approach transfer to a stronger base model?_
>
> \
> We respectfully point out that the performance gap stems from a difference in model capacity, not the baseline quality.
>
> - **Clarification on Model Size**: The scores you referenced (83.1 on VOC) are from **ViT-Giant (1B params)**. In contrast, our reported results in Table 2 are based on **ViT-Small (21M) and ViT-Base (86M), which are significantly smaller**. Given the 10x-50x difference in parameter count, the performance gap is expected. As such, we believe the numbers reported in our paper present a reliable comparison consistent with previous works.
>
> In the Dense SSL literature (e.g., Leopart[1], NeCo[2], CrIBO[3]), it is standard practice to evaluate methods on ViT-Small (21M) and ViT-Base (86M) architectures to ensure fair benchmarking of feature learning capabilities. We would like to emphasize that our reported numbers are fully consistent with the established state-of-the-art dense SSL approaches.
>
> However, to address your concern regarding the applicability of our approach to stronger base models, we conducted additional experiments using DINOv3.
>
> \
> **Table 1: Applying PaKA to DINOv3.** We evaluate DINOv3 with and without PaKA post-training, and observe consistent performance improvements on overclustering task.
> | **Method** | **Pretrained** | VOC (K=GT) | VOC (K=300) | VOC (K=500) | ADE20K (K=GT) | ADE20K  (K=300) | ADE20K  (K=500) |
>  |-----------|--------|-------|-------|-------|-------|-------|-------|
>   | DINOv3 | - | 16.1 | 55.2 | 60.4 | 9.9 | 13.3 | 16.6 |
>    | PaKA↑ | DINOv3 | **16.7** | **58.9** | **62.8** | **10.2** | **13.7** | **16.7** |
>
> \
> The results confirm that our method successfully improves DINOv3's dense understanding. Notably, this improvement is achieved with extremely **low training costs, requiring only 15 hours of training** on a single H100 GPU, which is a fraction of the cost compared to pretraining.
>
> [1] Leopart: Self-Supervised Learning of Object Parts for Semantic Segmentation, Ziegler et al., CVPR 2022
>
> [2] Near, far: Patch-ordering enhances vision foundation models' scene understanding, Pariza et al., ICLR 2025
>
> [3] CrIBo: Self-Supervised Learning via Cross-Image Object-Level Bootstrapping, Lebailly et al., ICLR 2024
>
>
> &nbsp;
> > **[Table 5a Performance]** _CKA is proposed as one of the main contributions in the paper but does not lead to a significant improvement in the linear probing protocol in Table 5a._
>
> \
> We appreciate the reviewer's detailed observation. While the linear probing scores in Table 5a are similar, we respectfully emphasize that the superiority of CKA lies in its robustness to structural misalignment and superior performance in fine-grained tasks.
>
> - **Beyond Linear Probing**: As shown in Table 5a, while linear probing scores are comparable, CKA significantly outperforms the Gram matrix method in Overclustering ($K=GT$) by +3.3%. This indicates that CKA is much more effective at capturing intrinsic, fine-grained semantic clusters which simple linear classifiers might not fully exploit.
> - **Analytical Contribution (Section 3.2)**: Our choice of CKA is not merely empirical but analytically grounded. As detailed in Section 3.2, we: (1) identified a "feature space gap" marked by geometric anisotropy between teacher and student distributions, and (2) demonstrated that CKA yields more stable training and faster convergence (Figure 2b) by aligning intrinsic structures, unlike the geometry-sensitive Gram matrix.
> - **Performance in Complex Tasks**: To further validate this, we conducted additional experiments on end-to-end fine-tuning with transformer decoder and semantic correspondence. In these structurally demanding tasks, CKA consistently outperforms the Gram matrix baseline.
>
> \
> **Table 2: CKA vs. Gram Matrix in Structurally Demanding Tasks.** We compare Gram matrix and CKA on End-to-End Segmentation (**EES**) using Pascal VOC and Sementic Correspondence (**SC**) using Spair-71k[4].
> | Method | EES (mIoU) | SC ($PCK_{image}$) | SC ($PCK_{point}$) |
> |--------|:-----------------------------:|:---------------------------:|:---------------------------:|
> | DINOv2 | 84.39 | 39.16 | 44.40 |
> | Gram | 83.94 | 41.09 | 46.09 |
> | CKA | **84.43** | **41.94** | **46.75** |
>
> [4] SPair-71k: A Large-scale Benchmark for Semantic Correspondence, Min et al., ICCV 2019

---

> > ### Author Response · Authors · 2025-11-26
> >
> > > **[Post-Training with DINO Loss and Gram Anchoring]** _The authors do not control for post-training on a different data distribution (COCO). What would be the performance of the DINO baseline, post-trained on COCO using the regular DINO loss and gram anchoring?_
> >
> > \
> > Thank you for the question. We would like to clarify that our **post-training is conducted on COCO, which is out-of-domain relative to the downstream evaluation datasets.** Therefore, the improvements are not a result of similarity between the training and evaluation datasets.
> >
> > \
> > Moreover, PaKA, which starts from Dinov2r backbone, also shows strong gains even when post-trained on ImageNet, which shares the same domain as the original DINOv2R backbone. In K=500, our method boosted Dinov2r's overclustering performance by +18.8%. This demonstrates that the improvement primarily comes from our loss design rather than dataset distribution.
> >
> > \
> > **Table 3: Effect of Data Distribution on PaKA.**  We compare post-training performance on ImageNet (in-domain) and COCO (out-of-domain), and evaluate overclustering performance on Pascal VOC. PaKA yields substantial gains in both settings, demonstrating that the improvement arises from the loss design rather than dataset similarity.
> > | **method** | Dataset   | Post-trained | K=GT | K=300 | K=500 |
> > |------------|-----------|--------------|------|-------|-------|
> > | DINOv2R    | -         | -            | 12.2 | 46.7  | 49.5  |
> > | PaKA       | ImageNet  | DINOv2R            | 16.9 | 65.3  | 68.3  |
> > | PaKA       | COCO      | DINOv2R            | 21.3 | 74.5  | 76.5  |
> >
> > \
> > +We will additionally evaluate standard DINO loss and gram anchoring on COCO to further isolate the effect of data distribution and include the results in the final version.
> >
> > &nbsp;
> > > **[Other Supervised Baselines & Non-linear Decoders]** _Other supervised baselines could be included, such as PESpatial or AM-RADIOv2.5, and the evaluation could go beyond the linear probing protocol_
> >
> > \
> > We respectfully exclude PESpatial and AM-RADIOv2.5 from our baselines for the following reasons:
> >
> > - **Training Paradigm**: These models utilize multi-teacher distillation from external oracles (e.g., SAM), whereas PaKA is strictly self-supervised (self-distillation).
> > - **Scope & Efficiency**: They aim to build foundation models via massive training, while PaKA focuses on efficient post-training refinement (14 GPU-hours).
> > - **Fairness**: We compare against relevant dense SSL methods (e.g., NeCo, CrIBO)  that operate under the same self-supervised constraints to ensure a fair assessment.
> >
> > \
> > Following the reviewer's suggestion, we conducted additional end-to-end segmentation experiments using a Transformer Decoder.
> >
> > \
> > The results confirm that PaKA consistently improves upon the DINOv2 baseline. While the performance gap is naturally narrower compared to linear probing due to the strong capacity of the Transformer decoder, PaKA still achieves the highest accuracy across both benchmarks, as detailed below:
> >
> > \
> > **Table 4: End-to-End segmentation experiments using a Transformer Decoder.**
> > | Method   | Pascal VOC (mIoU)  | ADE20K (mIoU)   |
> > |----------|:----:|:-----:|
> > | DINOv2   | 84.39 | 45.85 |
> > | Neco     | 83.03 | 45.85 |
> > | Paka     | **84.43** | **46.59** |

---

### Official Review · Reviewer_DF2M · 2025-10-29

**Soundness:** 2
**Presentation:** 3
**Contribution:** 2
**Rating:** 4
**Confidence:** 4

**Summary:**

This paper presents PaKA (Patch-level Kernel Alignment), a method for improving dense visual representations via self-supervised learning. Unlike clustering- or sorting-based approaches (e.g., iBOT, NeCo), PaKA proposes a non-parametric kernel-based alignment objective grounded in Centered Kernel Alignment (CKA). The method aligns the relational structure of patch embeddings between a student and teacher network, trained under an EMA framework.
Key design choices include:
- An CKA loss aligning pairwise patch similarities between student and teacher.
- A “clean teacher” (minimal augmentations) and high-overlap crops for consistent patch correspondence.
- Lightweight post-pretraining fine-tuning applicable to pretrained ViT encoders.
Empirically, PaKA achieves state-of-the-art performance on dense vision benchmarks (PASCAL VOC, COCO-Stuff, ADE20K) and demonstrates improved efficiency (–37% training time, –24% memory vs. NeCo). The paper is positioned as a generic refinement step rather than a full pretraining framework.

Summary of the review: The paper proposes PaKA, a simple and efficient refinement to dense self-supervised learning using a CKA-based objective that improves segmentation and depth performance while reducing compute and memory cost. The approach is practical and well-engineered, showing robustness and strong applicability to pretrained encoders. However, it lacks deeper conceptual novelty, mechanistic insight, and breadth of validation—key ablations, teacher analysis, and evaluations on classification, detection, or multimodal models are missing, and efficiency comparisons need clearer fairness. Overall, the work is technically sound and useful but remains primarily incremental rather than foundational, justifying a rating of 4 for solid engineering with limited conceptual advance.

Reproducibility Comment: Code provided is clean and structured with clear instructions. In the paper there is a clear provision of different evaluations and setups used. The main details for reproducibility are provided.

**Strengths:**

1) Simplicity: CKA provides a clean, non-parametric alternative to cluster-based dense SSL.
2) Empirical robustness: Consistent gains on dense segmentation and depth benchmarks.
3) Efficiency: Lower compute and memory cost than NeCo.
4) Practical relevance: A lightweight post-pretraining recipe applicable to strong pretrained encoders.

**Weaknesses:**

1. Batch-size dependence (missing ablation).
The authors acknowledge batch importance but show no study of how varying batch size affects convergence or downstream accuracy.

--> include 8–64 batch ablation for segmentation/hummingbird evaluation metrics.

2. Insufficient validation of “clean teacher.”
All evidence for the augmentation-free teacher is based on unsupervised segmentation, which is noisy.

--> Test the teacher-augmentation ablation using linear segmentation or Hummingbird-style stable probes that are less dependent on clustering variance.

3. Weak mechanistic motivation.
The paper demonstrates empirical gains but lacks an analysis explaining why the CKA objective enhances semantics (would also be interesting to see if it improves other metrics like object-detection as well and whether the motivation also provides insights for their task generalisability, therefore further explanation is expected ).

4. Limited understanding of PaKA's capabilities in global understanding.
PaKA’s effect on global understanding remains untested.

--> evaluate k-NN or linear classification on ImageNet vs. DINOv2R to ensure that dense refinement does not degrade global representations.

5. No exploration of VLM encoders (CLIP, SigLIP).
Given their prevalence, applying PaKA to CLIP or SigLIP vision encoders would clarify if the method benefits or harms text aligned vision models.

--> Report zero-shot classification, retrieval, segmentation performance, and hummingbird evaluation pre/post PaKA on them.

6. Finetuning evaluation limited to linear heads.
There are no non-linear decoders used for tasks such as segmentation (as well as depth estimation)

--> Evaluate with non-linear decoders (e.g.,Mask2Former) to ensure benefits persist in end-to-end settings.

7. Limited generalisation to non-dense tasks.
No other evaluations to simple segmentation, and depth estimation.

--> Evaluate with other benchmarks like object-detection (VITDET), Panoptic Segmentation (e.g., Mask2Former), since both PaKA and its predecessor seem very good in the hummingbird evaluation (fetching related patches) the question that is raised from my end is how good the features are also in key point matching (use the features from PaKA as feature descriptors to evaluate how good are they in key point matching eg on the HPatches Dataset [3]), and the Multiview feature consistency [1]

8. Dataset choice unexplained.
Why is COCO preferred for post-training instead of ImageNet or mixed-domain corpora?

--> A dataset ablation would clarify domain dependence.

9. Lacks of novelty to the creation of a new paradigm, seems more as an engineering investigation.
The method feels like a well-tuned refinement of existing self-distillation pipelines rather than a fundamentally new paradigm.

--> clarify more the position of PaKA ( is it an empirical improvement or a conceptual shift)

10. Computational (Execution and Memory) Cost Improvements are inconclusive
The reported efficiency gains (−37% time, −24% memory) may partly arise from engineering or implementation differences rather than inherent algorithmic simplicity. If PaKA employs xformers, torch compiled models, lighter augmentations, smaller intermediate tensors, mixed-precision optimization, or simply uses newer and more optimised libraries and/or has optimised their code further while NeCo’s public implementation does not, the comparison becomes uneven.

-->  (a) clarify the exact setup of execution comparisons, (b) since the code structure is very similar to NeCo provide a side by side comparison of computational costs (time and memory) on a per part of the code that you introduced eg (data processing, augmentations, forward pass, loss, etc),  (c) potentially average the computational costs over multiple runs to avoid any hardware noise,


[1] Banani, M. E., Raj, A., Maninis, K.-K., Kar, A., Li, Y., Rubinstein, M., … Jampani, V. (2024). Probing the 3D Awareness of Visual Foundation Models. arXiv [Cs.CV]. http://arxiv.org/abs/2404.08636

[2] Cheng, B., Misra, I., Schwing, A. G., Kirillov, A., & Girdhar, R. (2022). Masked-attention Mask Transformer for Universal Image Segmentation. arXiv [Cs.CV].  http://arxiv.org/abs/2112.01527

[3] Balntas, V., Lenc, K., Vedaldi, A., & Mikolajczyk, K. (2017). HPatches: A benchmark and evaluation of handcrafted and learned local descriptors. arXiv [Cs.CV].  http://arxiv.org/abs/1704.05939

**Questions:**

Questions for Authors are derived from the weaknesses:

1. Batch-size dependence
Could you include an ablation studying how varying batch size (e.g., 8–64) affects convergence, stability, and downstream accuracy?
This is particularly relevant since relational losses (like CKA) are sensitive to sample diversity and normalization across the batch.

2. Validation of the “clean teacher” design
Can you validate the effect of the augmentation-free teacher using more stable metrics, such as linear segmentation or Hummingbird-style evaluations, instead of unsupervised clustering-based segmentation, which is inherently noisy?
This would strengthen the causal link between the teacher’s augmentation level and representation quality.

3. Mechanistic motivation and conceptual clarity
Could you expand on why the CKA objective improves semantics?
Do you observe improved structure preservation or inter-class separation in the learned representation?
Would this mechanistic intuition also explain potential improvements in object detection or other structured tasks?

4. Global understanding and classification ability
How does PaKA influence global-level understanding?
Please evaluate on k-NN or linear classification (e.g., ImageNet-1k) compared to DINOv2R to ensure that dense refinement does not harm global semantic performance.

5. VLM encoder evaluation (CLIP, SigLIP)
Have you tested PaKA on vision encoders from multimodal models such as CLIP or SigLIP (with frozen text towers)?
Reporting zero-shot classification, retrieval, and segmentation performance (and optionally Hummingbird scores) before and after PaKA would clarify whether the method generalizes to text-aligned encoders without degrading cross-modal alignment.

6. Finetuning with non-linear decoders
Can you evaluate PaKA-pretrained encoders with decoder-based architectures (e.g., Mask2Former [2]) for segmentation and/or depth estimation?
This would show whether improvements persist in end-to-end fine-tuning beyond linear heads.

7. Generalisation to broader visual tasks
Could you test PaKA on additional dense and geometric benchmarks such as:
a. Object detection (e.g., ViTDet),
b. Panoptic segmentation (e.g., Mask2Former),
c. Keypoint matching using PaKA features as descriptors on HPatches [3], and
d. Multiview feature consistency or 3D awareness tests as in Banani et al. (CVPR 2024) [1]?
This would better establish the method’s versatility and geometric consistency.

8. Dataset choice for post-training
Why was COCO selected as the post-training dataset instead of ImageNet or a mixed-domain corpus?
Could you provide a small dataset ablation (e.g., ImageNet vs COCO) to demonstrate that PaKA’s gains are not domain-dependent?

9. Positioning and novelty of PaKA
How should readers interpret PaKA in the broader SSL landscape—do you consider it a conceptual advance or a well-engineered refinement of self-distillation pipelines?
Clarifying this positioning would help reviewers and readers understand its intended scope and contribution level.

10. Fairness of computational efficiency claims
The reported runtime and memory reductions (−37% time, −24% memory) might depend on implementation-specific optimizations.
Could you:
a) Clarify whether both PaKA and NeCo were benchmarked under identical optimization conditions (precision, compiler, libraries, data loading, etc.)?
b) Provide a side-by-side breakdown of time and memory usage per module (data loading, augmentations, forward pass, loss computation, etc.)?
c) Average timings over multiple runs to reduce hardware noise?
This would ensure the efficiency comparison is methodologically fair.

[1] Banani, M. E., Raj, A., Maninis, K.-K., Kar, A., Li, Y., Rubinstein, M., … Jampani, V. (2024). Probing the 3D Awareness of Visual Foundation Models. arXiv [Cs.CV]. http://arxiv.org/abs/2404.08636

[2] Cheng, B., Misra, I., Schwing, A. G., Kirillov, A., & Girdhar, R. (2022). Masked-attention Mask Transformer for Universal Image Segmentation. arXiv [Cs.CV].  http://arxiv.org/abs/2112.01527

[3] Balntas, V., Lenc, K., Vedaldi, A., & Mikolajczyk, K. (2017). HPatches: A benchmark and evaluation of handcrafted and learned local descriptors. arXiv [Cs.CV].  http://arxiv.org/abs/1704.05939

---

> ### Author Response · Authors · 2025-11-27
>
> We sincerely appreciate your constructive feedback and insightful questions. Your comments have made a tangible contribution to the quality of our manuscript. Specifically, inspired by your suggestion, we have introduced a new subsection, '5.3 Enhanced Semantic Discriminability and Matching' in the revised paper. We are grateful for your guidance in helping us strengthen the motivation of our work. Below, we address each of your points in detail.
>
> &nbsp;
> > **[Batch-size Ablation]** _include 8–64 batch ablation for segmentation/hummingbird evaluation metrics_
>
> \
> We have conducted a batch size ablation study. As shown in the table, we observe a consistent improvement in downstream performance as the batch size increases. We expect that the **performance could further improve with even larger batch sizes, given sufficient computational resources.**
>
> \
> **Table 1: Impact of Batch Size on Performance.** We conducted overclustering on Pascal VOC task with a range of granularities K: Ground-Truth (GT), 300, and 500.
>  |**Batch size** | K=21 | K=300 | K=500 |
> |--------|-------|-------|-------|
> | 8 | 14.2 | 62.2 | 65.5 |
> | 16 | 20.7 | 70.4 | 73.0 |
> | 32 | **21.3** | 72.8 | 75.0 |
> | 64 | 20.2 | **74.0** | **76.4** |
>
> &nbsp;
> > **[Weak Mechanistic Motivation]** _Mechanistic motivation and conceptual clarity Could you expand on why the CKA objective improves semantics? Do you observe improved structure preservation or inter-class separation in the learned representation? Would this mechanistic intuition also explain potential improvements in object detection or other structured tasks?_
>
> \
> We sincerely appreciate this insightful question. It prompted us to deeply investigate the geometric properties of the learned feature space, leading to significant improvements in our manuscript. To address your query regarding structure preservation and inter-class separation, we have added a new section (Section 5.3) with the following analyses:
>
> - **Qualitative Analysis**: We visualized the learned representations using t-SNE (Figure 5 in the revised paper). The results clearly reveal that PaKA produces a highly structured feature space. Crucially, PaKA effectively distinguishes between semantically similar and often confusing categories, such as **dogs versus horses**, where baseline methods frequently exhibit blurred boundaries.
> - **Quantitative Analysis**: To demonstrate that this separation translates to better dense semantics, we introduced the Semantic Correspondence benchmark. The results (Table 4 in the revised paper) show that PaKA effectively enhances the semantics of dense representations, particularly in establishing robust point-level correspondences.
>
> (The specific quantitative improvements on this benchmark are detailed in the following response.)
>
> &nbsp;
> > **[Global Understanding and Classification Ability]** _evaluate k-NN or linear classification on ImageNet vs. DINOv2R to ensure that dense refinement does not degrade global representations_
>
> \
> Thank you for the valuable suggestion. As you recommended, we conducted a k-NN classification experiment on ImageNet using the [CLS] token to evaluate global understanding, comparing PaKA to the DINOv2 baseline.
>
> The results indicate that PaKA's performance on this global task is slightly lower than that of the original DINOv2. We interpret this as a direct trade-off of our objective: while DINOv2 distills global information primarily into the specialized [CLS] token, PaKA's alignment objective encourages this context to be distributed across the individual patch tokens, thereby enhancing them for dense tasks at the expense of the [CLS] token's specialization.
>
> \
> **Table 2: Classification on ImageNet.** We evaluate global representation quality on ImageNet using both k-NN and linear probing with [CLS] token features.
>
> | Method | Metric | DINOv2 | NeCo | PaKA |
> | :---: | :---: | :---: | :---: | :---: |
> | **k=10** | Top-1 | 79.02 | 74.79 | 75.79 |
> |         | Top-5 | 91.62 | 89.12 | 89.70 |
> | **k=50** | Top-1 | 78.57 | 74.13 | 75.18 |
> |         | Top-5 | 93.96 | 91.76 | 92.46 |
> | **k=100**| Top-1 | 77.91 | 73.35 | 74.49 |
> |         | Top-5 | 94.29 | 92.06 | 92.74 |
> | **Linear**| Top-1 | 79.26 | 76.69 | 77.89 |
> |         | Top-5 | 94.77 | 93.52 | 94.16 |

---

> ### Author Response · Authors · 2025-11-27
>
> > **[Exploration of VLM Encoders]** _VLM encoder evaluation (CLIP, SigLIP) Have you tested PaKA on vision encoders from multimodal models such as CLIP or SigLIP (with frozen text towers)? Reporting zero-shot classification, retrieval, and segmentation performance (and optionally Hummingbird scores) before and after PaKA would clarify whether the method generalizes to text-aligned encoders without degrading cross-modal alignment._
>
> \
> Thank you for the valuable comment. Following your suggestion, we additionally applied PaKA to a Vision-Language Model, CLIP [1], by post-training its vision encoder.
>
> - **Improving Vision Understanding in CLIP**: As shown below, PaKA significantly improves the overclustering performance of CLIP, achieving substantial gains across all granularities (K=GT, K=300, K=500), compared to another post-training method, NeCo. This confirms that **our method generalizes well to non-DINO backbones and is not tied to a specific pretraining architecture.**
>
> \
> **Table 3: Applying PaKA to Vision-Language Model.** We conducted overclustering task with a range of granularities K, which is ground-truth, and 300 and 500.
> | **method** | Pretrained |  K=GT | K=300 | K=500 |
> |---|---|---|---|---|
> | CLIP   | - |  15.0 | 49.3  | 50.9  |
> | NeCo       | CLIP  |  14.2 | 52.2  | 55.7 |
> | PaKA       | CLIP  |  **16.6** | **54.3** | **56.9**  |
>
> &nbsp;
> - **Cross-Modal Alignment Preservation**: We further evaluated the post-trained vision encoder in a zero-shot segmentation setting with the pre-trained CLIP text encoder. Although PaKA does not explicitly optimize cross-modal alignment, only a **marginal drop** is observed, and the degradation is significantly smaller than that of NeCo. This shows that PaKA **preserves text–vision alignment much more effectively** while enhancing dense vision understanding.
>
> \
> **Table 4: Zero-Shot Segmentation in CLIP.** We evaluated the post-trained CLIP vision encoder with the original CLIP text encoder in a zero-shot segmentation setup across COCO-Stuff, ADE20K, and Pascal Context.
> | **Method** | Pretrained | **COCO-Stuff (mIoU)** | **COCO-Stuff (mAcc)** | **ADE20K (mIoU)** | **ADE20K (mAcc)** | **Pascal-Context (mIoU)** | **Pascal-Context (mAcc)** |
> |---|---|---|---|---|---|----|----|
> | CLIP | - | 14.5 | 32.8 | 9.1 | 25.1 | 21.9 | 44.0 |
> | NeCo | CLIP | 11.7 | 29.1 | 7.1 | 22.0 | 17.8 | 38.9 |
> | PaKA | CLIP | 14.1 | 31.8 | 8.4 | 23.3 | 21.6 | 43.6 |
>
>
> &nbsp;
> > **[Non-linear Decoders]** _Evaluate with non-linear decoders (e.g.,Mask2Former) to ensure benefits persist in end-to-end settings._
>
> \
> We appreciate the valuable suggestion to evaluate our method with non-linear decoders. To verify this, we conducted additional end-to-end segmentation experiments using a Transformer Decoder. Our decoder (2 transformer layers, 6 heads) prepends class embeddings to the input patches and attend jointly, after which the refined class embeddings are used to form the mask.
>
> The results confirm that PaKA consistently improves upon the DINOv2 baseline. While the performance gap is naturally narrower compared to linear probing due to the strong capacity of the Transformer decoder, PaKA still achieves the highest accuracy across multiple benchmarks, as detailed below:
>
> \
> **Table 5: End-to-End segmentation experiments using a Transformer Decoder.** We conducted end-to-end segmentation experiments using a Transformer Decoder, showing that PaKA outperforms the DINOv2 baseline on Pascal VOC, COCO-Stuff, and ADE20K.
> | Method| Pascal VOC (mIoU)  | COCO-Things (mIoU) | COCO-Stuff (mIoU) | ADE20K (mIoU) |
> |----------|:----:|:-----:|:----:|:-----:|
> | DINOv2   | 82.74 | **84.12**  | 61.72 | 45.95 |
> | NeCo     | 83.03 | 83.33 | 61.99 | 45.86 |
> | Paka     | **84.43** | 83.89 | **62.21** | **46.59** |

---

> ### Author Response · Authors · 2025-11-27
>
> > **[Generalisation to Broader Visual Tasks]** _Evaluate with other benchmarks like object-detection (VITDET), Panoptic Segmentation (e.g., Mask2Former), since both PaKA and its predecessor seem very good in the hummingbird evaluation (fetching related patches) the question that is raised from my end is how good the features are also in key point matching (use the features from PaKA as feature descriptors to evaluate how good are they in key point matching eg on the HPatches Dataset [3]), and the Multiview feature consistency_
>
> \
> We thank the reviewer for these valuable suggestions to broaden our evaluation. While we are currently working on expanding to Object Detection, we have prioritized evaluating our features on **Semantic Correspondence** (Section 5.3) and **Panoptic Segmentation** (Appendix B.2). We believe this task effectively addresses your question regarding feature quality and matching capabilities.
>
> &nbsp;
> - **Challenges in Semantic Correspondence**: Beyond standard keypoint matching (e.g., HPatches) which targets identical scenes or objects, semantic correspondence requires identifying semantically equivalent points across different instances of the same category (e.g., matching the "left ear" of two different dogs). This demands a deeper semantic understanding beyond low-level texture matching.
>
> \
> **Table 6: Semantic Correspondence.** We evaluated our features on semantic correspondence using SPair-71k, demonstrating their ability to match semantically equivalent points across different instances of the same category.
> | Method | PCK per Image (Spair-71k) | PCK per Point (Spair-71k) |
> |---|:---:|:---:|
> | DINOv2 | 39.16 | 44.40 |
> | PaKA |  41.94 | 46.75 |
>
> (The complete detailed results are provided in Section 5.3.)
>
> &nbsp;
> - **Challenges in Panoptic Segmentation**: Unlike standard semantic segmentation which treats the scene as a set of classes, panoptic segmentation requires a unified understanding of both countable instances ("things") and amorphous background regions ("stuff"). This demands a representation that can simultaneously maintain local instance discriminability and global semantic consistency, making it a rigorous test for the dense feature quality of the backbone.
>
> \
> **Table 7: Panoptic Segmentation.** We evaluated our features on panoptic segmentation using ADE20k. Metrics include overall $PQ$ and their variants:
> $PQ_{th}$ for _thing_ classes, and
> $PQ_{st}$ for _stuff_ classes.
> | Method | $PQ$ | $PQ_{th}$ | $PQ_{st}$ |
> |--------|:---:|:-----:|:-----:|
> | DINOv2R | 22.42 | 20.65 | 25.96 |
> | NeCo | 22.26 | 20.18 | 26.42 |
> | PaKA | **23.53** | **21.15** | **28.29** |
>
> (The complete detailed results are provided in Appendix B.2.)
>
> &nbsp;
> > **[Dataset Ablation]** _Dataset choice for post-training. Why was COCO selected as the post-training dataset instead of ImageNet or a mixed-domain corpus? Could you provide a small dataset ablation (e.g., ImageNet vs COCO) to demonstrate that PaKA’s gains are not domain-dependent?_
>
> \
> We additionally applied our method using ImageNet for post-training and reported the results in the Appendix B.10. As shown in the table, our method improves the performance of the DINOv2R backbone by up to +18.8% at K = 500, even though ImageNet is an object-centric dataset.
>
> \
> **Table 8: Post-training on In-Domain (ImageNet) vs Out-of-Domain dataset (COCO).**
> | **method** | Dataset  | Post-trained | K=GT | K=300 | K=500 |
> |----|---|---|---|---|---|
> | DINOv2R | - | -  | 12.2 | 46.7 | 49.5  |
> | PaKA | ImageNet | ✔ | 16.9 | 65.3 | 68.3 |
> | PaKA | COCO | ✔  | 21.3 | 74.5 | 76.5 |
>
> &nbsp;
> > **[Positioning and Novelty of PaKA]** _How should readers interpret PaKA in the broader SSL landscape—do you consider it a conceptual advance or a well-engineered refinement of self-distillation pipelines? Clarifying this positioning would help reviewers and readers understand its intended scope and contribution level._
>
> \
> We respectfully clarify that PaKA represents a **conceptual shift** beyond an engineering investigation. Our work proposes a fundamental paradigm shift in dense self-supervised learning: moving from parametric, constrained algorithms (e.g., clustering or sorting) to a **non-parametric, assumption-free kernel-based approach**.
>
> \
> This shift significantly **reduces the complexity** of fine-grained learning while improving performance across various downstream tasks. To the best of our knowledge, we are the first to introduce a kernel-based learning method to dense SSL.
>
> \
> Furthermore, we identify a critical "structural misalignment" between teacher and student feature spaces (as shown in Figure 2a ) that renders standard alignment methods unstable. Consequently, our approach is not a mere tuning trick, but a **statistically grounded solution** to the **distribution discrepancy problem** inherent in dense representation learning. To complement this conceptual basis, we have leveraged your insightful feedback to reinforce the motivation of our framework.

---

> ### Author Response · Authors · 2025-11-27
>
> > **[Fairness of Computational Efficiency]** _The reported runtime and memory reductions (−37% time, −24% memory) might depend on implementation-specific optimizations. Could you: a) Clarify whether both PaKA and NeCo were benchmarked under identical optimization conditions (precision, compiler, libraries, data loading, etc.)? b) Provide a side-by-side breakdown of time and memory usage per module (data loading, augmentations, forward pass, loss computation, etc.)? c) Average timings over multiple runs to reduce hardware noise? This would ensure the efficiency comparison is methodologically fair._
>
> \
> We appreciate this question, as it allows us to clearly explain the algorithmic reasons behind the performance gap.
>
> &nbsp;
> - **Clarification on Experimental Fairness**: First, we explicitly confirm that **the results reported in our main paper** were measured under strictly identical environments and conditions. Both PaKA and NeCo used the same hardware, compiler settings, precision, and library versions. Crucially, no method-specific low-level optimizations (like fused kernels) were applied to PaKA that were not applied to NeCo. The reported efficiency gains are **purely algorithmic**.
>
> \
> Following your suggestion, we conducted an additional granular profiling to provide a side-by-side breakdown. Tracking the precise memory consumption of specific modules throughout the entire training process is challenging. Therefore, we measured the average GPU memory usage while performing the task. To reduce hardware noise, we averaged the values measured over 1000 steps during training.
>
> - Note on Hardware: While our main paper results were benchmarked on NVIDIA H100 GPUs, this specific breakdown was performed on a single RTX 3090 GPU with a batch size of 10 due to current resource constraints.
>
> \
> **Table 9: Breakdown of runtime and memory usage for PaKA and NeCo.**
> | module | PaKA Time (ms) | NeCo Time (ms) | PaKA Memory (MB) | NeCo Memory (MB) |
> | :--- | :--- | :--- | :--- | :--- |
> | Augmentation | 168.1 | 107.1 | - | - |
> | Teacher Forward | 282.9 | 282.3 | 1777.5 | 1695.5 |
> | Student Forward | 301.9 | 298.9 | 17725.5 | 17612.2 |
> | Loss Calculation | 12.8 | 234.4 | 17807.4 | 19557.1 |
> | Backward Pass | 739.8 | 901.4 | 17951.0 | 19509.8 |
> | Optimizer Step | 1355.3 | 1736.3 | 1048.9 | 966.8 |
>
> \
> The experimental results clearly indicate that the primary computational overhead stems from the Loss Calculation, Backward Pass, and Optimizer Step. Most notably, PaKA (12.8ms) is approximately **18x** faster than NeCo (234.4ms) during loss calculation. This efficiency gap naturally propagates to the backward pass and weight update stages.
>
> \
> The reason for this is clear from an **algorithmic perspective**. NeCo's differentiable sorting requires iteratively computing and multiplying a chain of soft permutation matrices to approximate ranking, which creates a deep computational graph that is expensive to backpropagate through. In contrast, PaKA's CKA loss relies on a fixed sequence of standard matrix operations (Gram matrix construction and Frobenius norm). These are **closed-form calculations** with a shallow computational graph, allowing for **significantly faster computation and efficient memory use**.

---

### Official Review · Reviewer_qWgt · 2025-10-30

**Soundness:** 3
**Presentation:** 3
**Contribution:** 3
**Rating:** 6
**Confidence:** 4

**Summary:**

This paper presents a novel post-(pre)training method for improving the low-level quality of self-supervised learning representations, and thus increase the downstream performance on tasks such as segmentation or overclustering. Experimental results show that applying on a DINOv2 model leads to significant improvements on benchmarks such as Pascal VOC, COCO, or ADE20k.

**Strengths:**

- The presented method is well-motivated, in particular the need for a non-parametric local-matching approach in dense self-supervised learning, and matching gram matrices on locally aggregated patch-level representations. The Centered Kernel Alignment (CKA) approach is also well motivated as opposed to using the actual gram matrix.

- The paper is very clear and well-written, the proposed method is correctly formalized and the figures go straight to the point and help the understanding.

- The evaluation setup is strong and makes the results convincing. Large-scale datasets are used (COCO, ADE20k), against strong baselines (DINOv2R), with meaningful evaluation protocol (linear probing).

**Weaknesses:**

- The presented method should be agnostic to any pretraining method. It would have been great to show this by post-training on more than a single model, right now there is no way of saying if this post-training is specific to DINO or if it works in the general case.

- The specific backbone used for most of the experiment is not specified in the main text or in the figures captions. I think this is a very important detail for practitioners that should be highlighted.

- Where does DINOv2R line performance come from in Table 1,2,3,4 ? Is it your reproduction or from another paper ? I can’t see these results on the original paper. This is why specifying the backbone is important, otherwise comparisons are meaningless.
Why do you compare your results to this specific version of DINO ? Why not DINOv2 or v3 ?

- I don’t think Section 4 is relevant, at least it should as much importance because these findings on data augmentation are very specific to the particular method that is proposed, and particular to the fact that this is post-training and not pre-training. The writing might be a bit misleading “Motivated by the limitations of this inherited augmentation paradigm” makes it believe that this is a new go-to augmentation strategy.

- There are a number of small presentation issues:

- L851 typo “evalutaions”

- L23 typo “Kernal”

- “For a fair comparison, we also post-trained same backbone model by NeCo (Pariza et al., 2025),” This is confusing , do you post-train from neco, which is pretrained from dinov2r according to the Table 1 ?

- Line 41 “another line of research (Lebailly et al., 2024; Pariza et al., 2025; Stegmüller et al., 2023; Ziegler & Asano, 2022) has focused on dense representation learning via self-distillation” DINOv2 and Ibot can also be considered as self-distillation methods

- Some figures are blurry, make sure to use .pdf instead of .png or .jpeg

- “while reducing computation by 37% and memory usage by 24% compared to prior methods” which one ? This needs to be precise and only can compare to post-training techniques. The answer is actually in section 5.4: “more memory-efficient compared to NeCo” it should be mentioned above.

**Questions:**

The method is used as a post-(pre)training refinement stage. Have you thought about making it a pretraining method ?

---

> ### Author Response · Authors · 2025-11-26
>
> We sincerely appreciate your feedback and insightful questions. We were especially grateful for the way you engaged with the details of our work and offered concrete points for improvement. Below, we address each of the points from your review.
>
> &nbsp;
> > **[Post-training Backbone]** _"The presented method should be agnostic to any pretraining method. It would have been great to show this by post-training on more than a single model, right now there is no way of saying if this post-training is specific to DINO or if it works in the general case."_
>
> \
> Thank you for the valuable comment. We agree that demonstrating generality beyond a single pretrained backbone is important. To verify that our post-training method is not specific to DINO, we additionally applied PaKA to a Vision-Language Model, CLIP [1], by post-training its vision encoder.
>
> As shown below, PaKA significantly improves the overclustering performance of CLIP, achieving substantial gains across all settings (K=GT, K=300, K=500), compared to another post-training method, NeCo. This shows that **our method generalizes well to non-DINO backbones and is not tied to a specific pretraining architecture.**
>
> \
> **Table 1: Applying PaKA to Vision-Language Model.** We conducted overclustering task with a range of granularities K: the ground-truth number of classes (K=GT), 300, and 500.
> | **method** | Pretrained |  K=GT | K=300 | K=500 |
> |------------|-----------|--------------|------|-------|
> | CLIP   | -         |  15.0 | 49.3  | 50.9  |
> | NeCo       | CLIP  |  14.2 | 52.2  | 55.7 |
> | PaKA       | CLIP  |  **16.6** | **54.3** | **56.9**  |
>
> \
> These results show that PaKA can effectively improve representations across a wide range of pretrained models, including both self-supervised approaches and vision-language models.
>
>
> [1] Learning Transferable Visual Models From Natural Language Supervision, Radford et al., ICML 2024
>
> &nbsp;
> > **[Details of Backbone]** _The specific backbone used for most of the experiment is not specified in the main text or in the figures captions. I think this is a very important detail for practitioners that should be highlighted._
>
> \
> The backbone and implementation details are described in Section 5.1 of the paper. Our model is post-trained from the DINOv2R (ViT-S/4) checkpoint [2] without registers. We agree that highlighting the backbone in figure captions will improve clarity, and we have added this information below the Figure 1.
>
>
> [2] Vision Transformers Need Registers, Darcet et al., ICLR 2024
>
> &nbsp;
> > **[Choice of DINOv2R Backbone]** _Where does DINOv2R line performance come from in Table 1,2,3,4 ? Is it your reproduction or from another paper ? I can’t see these results on the original paper. This is why specifying the backbone is important, otherwise comparisons are meaningless. Why do you compare your results to this specific version of DINO ? Why not DINOv2 or v3 ?_
>
> \
> The DINOv2R performance reported in Tables 1–4 comes directly from the baseline paper [3]. Following your suggestion, we have further evaluated our method on DINOv2 and DINOv3 to demonstrate the generalizability of our approach. Specifically, we post-trained these backbones with our method and reported results on overclustering. The experimental results show that our method consistently improves performance on both DINOv2 and DINOv3, confirming that the effectiveness of our approach is not tied to a specific backbone.
>
> \
> **Table 2: Applying PaKA to other DINO Backbones.** We conducted overclustering task with K = GT, 300, and 500. PaKA shows consistent performance gains when applied to different DINO backbones.
> | **Method** | **Pretrained** | VOC (K=GT) | VOC (K=300) | VOC (K=500) | ADE20K (K=GT) | ADE20K  (K=300) | ADE20K  (K=500) |
>  |-----------|--------|-------|-------|-------|-------|-------|-------|
> | DINOv2 | - | 13.8 | 47.6 | 54.2 | 7.9 | 10.2| 13.0 |
>  | PaKA↑ | Dinov2 | 12.4 | 55.5 | 59.8 | 10.0 | 13.8 | 17.0 |
>   | DINOv3 | - | 16.1 | 55.2 | 60.4 | 9.9 | 13.3 | 16.6 |
>   | NeCo↓| DINOv3 | 12.7 | 50.0 | 55.0 | 8.8 | 11.5 | 14.1 |
>    | PaKA↑ | DINOv3 | 16.7 | 58.9 | 62.8 | 10.2 | 13.7 | 16.7 |
>    | PaKA↑| DINOv2r | **21.3** | **74.5** | **76.5** | **12.1** | **17.4** | **21.4** |
>
> \
> **While PaKA consistently improves performance across all DINO backbones, the DINOv2R backbone achieves the best overall results when combined with our method.** Therefore, we chose DINOv2R as the primary backbone due to its stronger baseline performance and greater improvement under our method.
>
> [3] Near, far: Patch-ordering enhances vision foundation models' scene understanding, Pariza et al., ICLR 2025

---

> > ### Author Response · Authors · 2025-11-26
> >
> > > **[Scope of Augmentation: Specific to Post-Training]** _I don’t think Section 4 is relevant, at least it should as much importance because these findings on data augmentation are very specific to the particular method that is proposed, and particular to the fact that this is post-training and not pre-training. The writing might be a bit misleading “Motivated by the limitations of this inherited augmentation paradigm” makes it believe that this is a new go-to augmentation strategy._
> >
> > \
> > As the reviewer suggested, we revised the section to explicitly state that the limitation arises when existing augmentation strategies are applied exclusively during post-training. Importantly, our ablation results confirm that our method generalizes to other post-training frameworks like NeCo, showing that it is loss-agnostic rather than specific to CKA.
> >
> > Although our approach is indeed confined to post-training, it is an initial effort to identify and validate suitable augmentation methods within the domain of dense self-supervised learning.
> >
> > &nbsp;
> > > **[Minor Editorial Issues]** _There are a number of small presentation issues:_
> > _L851 typo “evalutaions” / L123 typo “Kernal”_
> >
> > \
> > We have corrected the typos pointed out by the reviewer. We truly appreciate your careful and thoughtful review of our manuscript.
> >
> > &nbsp;
> > > **[Clarity of Backbone initialization]** _“For a fair comparison, we also post-trained same backbone model by NeCo (Pariza et al., 2025),” This is confusing , do you post-train from neco, which is pretrained from dinov2r according to the Table 1 ?_
> >
> > \
> > To clarify, as stated in Section 5.1, our method is post-trained from the DINOv2R backbone, not from a NeCo checkpoint. Likewise, the NeCo baseline is also a post-training method that starts from the same DINOv2R checkpoint. For consistency, we reproduced NeCo under this setting **to ensure a fair comparison between post-training methods on an identical backbone (DINOv2R).**
> >
> > Therefore, both our method and NeCo start from the same pretrained DINOv2R checkpoint, which allows us to compare only the effectiveness of the post-training strategies themselves.
> >
> > &nbsp;
> > > **[Clarifying DINOv2 / iBOT]** _Line 41 “another line of research (Lebailly et al., 2024; Pariza et al., 2025; Stegmüller et al., 2023; Ziegler & Asano, 2022) has focused on dense representation learning via self-distillation” DINOv2 and Ibot can also be considered as self-distillation methods_
> >
> > \
> > We agree with your point regarding the classification of DINOv2 and iBOT. Accordingly, we have revised the text to distinguish the two lines of research based on their specific objectives (reconstruction vs. feature alignment) rather than the self-distillation framework. We truly appreciate your careful and thoughtful review.
> >
> > &nbsp;
> > > **[Replacement of Figures]** _Some figures are blurry, make sure to use .pdf instead of .png or .jpeg_
> >
> > \
> > We replaced the blurry Figures 2(a) and 2(b) with higher-quality PDF versions instead of PNG. We truly appreciate your careful and thoughtful review of our manuscript.
> >
> > &nbsp;
> > > **[Clarifying Comparison Target for Efficiency Claim]** _“while reducing computation by 37% and memory usage by 24% compared to prior methods” which one ? This needs to be precise and only can compare to post-training techniques. The answer is actually in section 5.4: “more memory-efficient compared to NeCo” it should be mentioned above._
> >
> > \
> > Thank you for pointing this out. We agree that the claim needs to be more precise and restricted to comparisons with post-training techniques. We will revise the sentence to explicitly state that the 37% reduction in computation and 24% reduction in memory usage are measured **with respect to NeCo** in 1. Introduction.
> >
> > &nbsp;
> > > **[Applicability for Pretraining method]** _The method is used as a post-(pre)training refinement stage. Have you thought about making it a pretraining method ?_
> >
> > &nbsp;
> > - **Designed for Patch-level Refinements**: As you suggested, we experimented with applying our loss as a pretraining objective from random initialization, but the performance collapsed. We believe this is because our objective does not involve the [CLS] token during training. The [CLS] token plays a crucial role in learning global semantic structure in standard SSL frameworks such as DINO. Instead, PaKA is designed to refine patch-level representations (via [PATCH] tokens) after the global semantics has already emerged through standard SSL pretraining.
> >
> > &nbsp;
> >
> > We acknowledge that a hybrid strategy—combining a global SSL objective (e.g., DINO loss via [CLS] tokens) with our patch-level refinement objective (via [PATCH] tokens), similar to CrIBO [4] and could be a useful direction. We appreciate the insightful suggestion and plan to explore this approach in future work.
> >
> > [4] CrIBo: Self-Supervised Learning via Cross-Image Object-Level Bootstrapping, Lebailly et al., ICLR 2024

---

### Author Response · Authors · 2025-11-27
**Common Response to All Reviewers**

We are deeply grateful for the time and effort the reviewers have dedicated to evaluating our paper. Your detailed feedback has provided us with valuable perspectives to strengthen our work.

Before addressing specific questions, we would like to reiterate the core motivation and distinct contributions of PaKA to clarify its position within the field of dense self-supervised learning (SSL).

&nbsp;
1. Motivation
- **From Parametric to Non-Parametric**: Our work marks a fundamental shift from parametric approaches—which rely on constrained algorithms like clustering (e.g., Croc[1], CrIBO[2]) or sorting (e.g., NeCo[3])—to a **non-parametric, assumption-free kernel-based approach**. (Section 2.2)
- **Capturing Holistic Structure**: Instead of imposing artificial constraints (e.g., number of clusters or ranking steepness), we utilize a kernel-based objective to capture the holistic similarity structure of high-dimensional feature distributions effectively. (Section 3.2)
- **Addressing Structural Misalignment (CKA)**: We address the critical 'structural misalignment' between teacher and student spaces (Figure 2a) by employing CKA. This aligns intrinsic statistical dependence, ensuring robustness despite distributional discrepancies. (Figure 2b)

&nbsp;
2. Key Contributions & Distinctiveness
- **Novelty**: PaKA is, to our knowledge, the first non-parametric kernel-based approach applied to dense self-supervised learning.
- **Simplicity (No Auxiliaries)**: We completely remove the reliance on memory banks, clustering, or sorting algorithms. This eliminates the hyperparameter sensitivity and training complexity inherent in previous methods (NeCo[3], CrIBO[2], etc.).
- **Efficiency & Performance**: PaKA offers a lightweight post-training stage that is computationally efficient yet consistently outperforms prior state-of-the-art methods on dense self-supervised learning.

\
Thank you again for your dedication and insightful comments. We remain fully open to further discussions and would be happy to address any remaining questions during the rebuttal period.

\
[1] CrOC: Cross-View Online Clustering for Dense Visual Representation Learning, Stegmüller et al., CVPR 2023

[2] CrIBo: Self-Supervised Learning via Cross-Image Object-Level Bootstrapping, Lebailly et al., ICLR 2024

[3] Near, far: Patch-ordering enhances vision foundation models' scene understanding, Pariza et al., ICLR 2025

---

### Author Response · Authors · 2025-12-03
**Summary of Discussion and Revisions for Area Chair**

Dear Reviewers and Area Chair,

We extend our sincere gratitude to the reviewers for their constructive feedback, which has guided us in improving both the quality and clarity of our submission. Through the rebuttal, we have addressed the reviewers' concerns with extensive new experiments and clearer articulation of our contributions.

To assist the Area Chair in their final assessment, we summarize the primary discussions and clarifications below.

&nbsp;
### Summary of Recognized Strengths
We appreciate that the reviewers commended the clear motivation behind our non-parametric dense post-training approach, describing the CKA-based objective as **"clean"**, **"well-motivated"**, and **"sound"** (qWgt, DF2M). Furthermore, the **"strong evaluation setup"** and **"consistent gains"** on large-scale dense benchmarks were widely acknowledged as evidence of our method’s robustness (qWgt, DF2M).

&nbsp;
### Key Discussions and Clarifications

The primary discussions revolved around the following concerns:
* (1) Generality Across Backbones,
* (2) Mechanistic Insight: Structure & Correspondence,
* (3) Generalisation to Broader Visual Tasks, and
* (4) Validity of Baseline Comparisons

**1. Generality Across Backbones (qWgt, DF2M)**

Reviewer qWgt asked whether PaKA is specific to DINOv2 or applicable to a broader set of backbones. To address this, we applied our post-training method not only to DINOv2R but also to DINOv2 [1], DINOv3 [2], and the vision–language model CLIP [3]. PaKA **consistently improved dense performance** across all these models. In particular, it efficiently enhanced CLIP’s dense understanding while **preserving text–vision alignment** and **further strengthened the highly competitive DINOv3 baseline**. (Appendix B.7, B.8).

**2. Mechanistic Insight: Structure & Correspondence (DF2M)**

Reviewer DF2M requested a deeper analysis of **_why_ and _how_ the CKA objective contributes** to semantic enhancement. In response, we first **refined the motivation and structural analysis** presented in our original submission, and provided these clarifications in the common response. To provide further empirical evidence, we analyzed the learned feature space and incorporated **a new section (Section 5.3)** into the revised manuscript. This section presents t-SNE visualizations that reveal **clear inter-class separation**, along with quantitative results on the challenging **Semantic Correspondence task**, demonstrating the method's effectiveness in preserving semantic structure.

**3. Generalisation to Broader Visual Tasks (DF2M)**

In response to Reviewer DF2M’s suggestion to broaden the experimental scope and examine how well PaKA can **generalize across different tasks**, we conducted a range of additional evaluations.

Our results demonstrate that PaKA:

**(1) Global Understanding:** robustly preserves global understanding capabilities in ImageNet classification (Appendix B.10);

**(2) Complex Dense Prediction:** excels in challenging structural tasks, as evidenced by performance in Semantic Correspondence (Section 5.3) and Panoptic Segmentation (Appendix B.2);

**(3) CLIP Zero-shot Segmentation:** it enhances CLIP's dense prediction capabilities while preserving text-vision alignment (Appendix B.7); and

**(4) Segmentation with a Non-linear Decoder:** maintains consistent improvements even within non-linear regimes (Appendix B.4).

These comprehensive findings strongly reinforce the motivation and versatility of our approach beyond standard benchmarks.

**4. Validity of Baseline Comparisons (CCN1)**

Reviewer CCN1 questioned the strength of our baselines, citing a performance gap compared to DINOv3 [2]. However, this concern arose from a fundamental misunderstanding: the reviewer compared our **ViT-Small/Base (21M/86M)** results against the **ViT-Giant (1B)** performance reported in the reference paper [2]. We clarified that the observed gap is a natural consequence of the **10x-50x difference in model capacity**, not a weakness of our baseline. By highlighting that our experimental setup strictly adheres to the standard Dense SSL literature, we clarified that the reviewer’s "main question" was based on an invalid comparison.

\
Our work marks a **fundamental shift** in dense self-supervised learning, moving from parametric approaches reliant on complex constraints like clustering or sorting to a **non-parametric, assumption-free kernel-based framework**. We believe this approach offers a simpler, statistically grounded foundation that not only eliminates hyperparameter sensitivity but also achieves **state-of-the-art performance with remarkable efficiency**.

\
Best regards,

Authors

\
**References**

[1] DINOv2: Learning Robust Visual Features without Supervision, Oquab et al., 2023.

[2] DINOv3, Siméoni et al., 2025.

[3] Learning Transferable Visual Models From Natural Language Supervision, Radford et al., ICML 2021.

---

### Meta-Review · Area_Chair_CCXQ · 2026-01-02

**Summary:**

This paper presents PaKA (Patch-level Kernel Alignment), a post-training method to improve the dense visual representations on top of pretrained ones for dense CV benchmarks  (e.g., PASCAL VOC, COCO-Stuff, ADE20K). PaKA proposes a non-parametric kernel-based alignment objective grounded in Centered Kernel Alignment (CKA). The method aligns the relational structure of patch embeddings between a student and teacher network, trained under an EMA framework. The author showed efficacy of proposed approaches by applying PaKA to a DINOv2 model, which yields significant improvements on dense CV benchmarks.

Reviewers found the paper is of following strength
* The work proposed a lightweight post-pretraining recipe applicable to pretrained encoders, which can be practically useful.
* The paper select reasonable benchmarks to demonstrate the effectiveness of proposed method. The evaluation setup is strong and makes the results convincing. Consistent gains can be observed on dense segmentation and depth benchmarks.
* The paper is very clear and well-written.

**Reviewer Concerns:**

We also found the additional explanation from authors during rebuttal are helpful (e.g., evaluation on more datasets and models including DINOv3 and CLIP, explaining more methodology details and insights). These are very helpful experiments/discussion and will strengthen this work.

Although this submission is of reasonable quality and contribution to science community, the reviewers found impact of this work is at the borderline compared to a typical ICLR paper. The work is technically sound but primarily incremental rather than foundational - it's an augmentation of existing pretrained representations for dense vision capabilities. It doesn't solve the fundamental problem where the dense and low level information is not modeled properly given the prevailing of representation learning with hidden layer as learning targets. To broaden impact so that general ICLR audience would find interested, we encourage the authors to focus on attacking the problems at pretraining stage, or significantly expand the scope of current evaluation (e.g., applying PaKA to more representations, tasks, and thorough analysis). Even though the authors add more experiments during rebuttals to address reviewers' feedback and validate their claims, the reviewers believe the rebuttal phase is too short so only small amendment can be added. We believe a systematic design of evaluation to cover  more tasks and models would great increase the value of this work. Thus, we encourage the authors to resubmit to a later conference.

**Reviewer Scores:**

I would expect the reviewers may slightly increase scores during the rebuttal with the additional experiments and discussion. However, with the concerns discussed above in details about impact, there is no clear distinction that would likely convince reviewers to accept this paper.

---

### Decision · Program_Chairs · 2026-01-26

Reject